# Present-day mass loss rates are a precursor for West Antarctic Ice Sheet collapse

Tim van den Akker[1], William H. Lipscomb[2], Gunter R. Leguy[2], Jorjo Bernales[3], Constantijn J. Berends[1], Willem Jan van de Berg[1], Roderik S.W. van de Wal[1,4]

[1]Institute for Marine and Atmospheric Research Utrecht, Utrecht University, Netherlands

[2]Climate and Global Dynamics Laboratory, NSF National Center for Atmospheric Research, Boulder, CO, USA

[3]Danish Meteorological Institute, Copenhagen, Denmark

[4]Department of Physical Geography, Utrecht University, Netherlands

*Correspondence to*: Tim van den Akker (t.vandenakker@uu.nl)

**Abstract.** Observations of recent mass loss rates of the West Antarctic Ice Sheet (WAIS) raise concerns about its stability since a collapse would increase global sea levels by several meters. Future projections of these mass loss trends are often estimated using numerical ice sheet models, and recent studies have highlighted the need for models to be benchmarked against present-day, observed mass change rates. Here, we present an improved initialization method that optimizes local agreement not only with observations of ice thickness and surface velocity, but also with satellite-based estimates of mass change rates. This is achieved by a combination of tuned thermal forcing under the floating ice shelves and friction under the ice sheet. Starting from this improved present-day state, we generate an ensemble of future simulations of Antarctic mass change by varying model physical choices and parameter values while fixing the climate forcing at present-day values. The dynamical response shows slow grounding line retreat over several centuries, followed by a phase of rapid mass loss over about 200 years with a consistent rate of ~3 mm GMSL/yr. We find that for all ensemble members, the Thwaites and Pine Island glaciers collapse. Our results imply that present-day ocean thermal forcing, if held constant over multiple centuries, may be sufficient to deglaciate large parts of the WAIS, raising global mean sea level by at least a meter.

## Introduction

The projected Antarctic contribution to global mean sea level (GMSL) rise ranges from 0.03 m (SSP1-1.9, low end of the likely range) to 0.34 m (SSP5-8.5, high end of the likely range) in 2100 (Fox-Kemper et al., 2021). Dynamical processes are not expected to accelerate or decelerate ice mass change significantly before 2100 (Van De Wal et al., 2022). After 2100, however, the GMSL driven by nonlinear processes is highly uncertain, possibly accelerating GMSL rise significantly (Fox-Kemper et al., 2021; Payne et al., 2021).

One such process is the Marine Ice Sheet Instability (MISI; see Schoof (2012, 2007) and Durand et al. (2009), which could drive a mostly dynamical irreversible retreat (i.e. the glacier will not return to its present grounding line position when the forcing is removed) of marine-terminating glaciers. Robel et al. (2019) argued that the mere possibility of MISI amplifies uncertainty in sea level rise projections from ice sheet models. MISI can occur for parts of the ice sheet where outlet glaciers terminate on a retrograde-sloping, submerged bedrock, due to a positive feedback between the flux across the grounding line

(GL, the transition line between a grounded ice sheet and a floating ice shelf) and the ice thickness at the GL. Thwaites Glacier (TG) and Pine Island Glacier (PIG), in the Amundsen Sea region of the West Antarctic Ice Sheet (WAIS), are located in such regions and may therefore be vulnerable to MISI.

Currently, the Thwaites GL retreats by up to 1.2 km per year on the eastern shelf and up to 0.9 km per year along the western margin (Milillo et al., 2019), and the GL of PIG retreated 31 km in 20 years (Rignot et al., 2014). Warm Circumpolar Deep

Water (CDW) and related high basal melt rates have been observed under Pine Island Glacier (Jacobs et al., 2011a) since the early 1990s, and studies show that CDW intrusions have coincided with observed PIG speedup (Thoma et al., 2008; Jenkins et al., 2016). As a result, the Amundsen Sea Embayment (ASE, which contains TG and PIG) is currently the largest contributor to Antarctic Ice Sheet (AIS) mass loss (Smith et al., 2020; Rignot et al., 2019). Further retreat of PIG and TG would increase ice drainage from these basins into the ocean, eventually triggering the collapse (i.e. relatively fast,

potentially irreversible ungrounding and ice loss) of a significant part of the WAIS over several centuries.

Several modelling studies have assessed the potential for ASE collapse. One study (Joughin et al., 2014) argued that under present-day melt rates, TG might already be on a trajectory toward accelerated retreat; moderate retreat in this century will likely be followed by a phase of rapid collapse beginning in the next 200 to 900 years. Another study (Favier et al., 2014) used three ice sheet models to show that PIG is now undergoing a climate-driven 40-km retreat, but they made no

projections on longer time scales. Both studies mention MISI as the main driver of retreat, but the retreat might be reversed by sufficient ocean cooling (Favier et al., 2014). A number of recent studies suggest that TG and PIG are unstable under the current climate and could collapse on a timescale up to 2000 years (Garbe et al., 2020; Golledge et al., 2021; Reese et al., 2023; Coulon et al., 2024). Lipscomb et al. (2021) projected accelerated retreat leading to a collapse only when ocean thermal forcing increases by 1–2 K relative to a preindustrial or 20th century equilibrium. However, there are few historical

observations of Southern Ocean temperatures, so it is unknown how much the ASE has warmed in the past century, and if the warming needed to drive such a retreat has already happened.

Studying ASE glacier stability requires century-scale model simulations, which contain significant uncertainties. Seroussi et al. (2024) found that potential AIS contributions to GMSL rise dramatically after 2100, with large variations across ice sheet models. According to Aschwanden et al. (2021), uncertainties in ice sheet modelling arise from four main sources:

suboptimal model initialization, incomplete understanding of physical processes, numerical model uncertainty, and

uncertainty in the climate forcing. Some ice sheet models are initialized using a long spin-up to steady state. Since they are initialized to a stable ice sheet without model drift, they do not represent the observed mass change of recent decades. A good representation of recent mass changes is essential for reliable projections, as these changes are likely to continue or accelerate.

Broadly, two methods are used to initialize ice sheet models: data assimilation and spin-up. In the first, data assimilation methods (Larour et al., 2012; Gudmundsson et al., 2012; Cornford et al., 2015; Hoffman et al., 2018; Bradley and Arthern, 2021) capture conditions at a certain time. For example, based on the observed ice thickness, the ice surface velocities are calculated and compared to observations. Uncertain parameters are tuned to minimize a cost function based on the difference between observed and modelled quantities until the velocities converge to a state close to observations. This has the

advantage that the resulting ice sheet matches the observed thickness and is ideally close to observed velocities at the start of a forward run, without necessarily being in steady state.  However, the resulting dynamic state might not replicate the observed mass change rates. The simulation of recent mass change can be improved  where annual or subannual observations are available, by doing a transient calibration as suggested by Goldberg et al. (2015). Also, the modelled mass change rates can be added to the cost function, as was done for example by Rosier et al. (2021) to minimize model drift.

Alternatively, Bett et al. (2023) used the ice sheet model WAVI (Arthern et al., 2015) and added the difference between observed and modeled mass change rates to the cost function, using observed mass change rates from Smith et al. (2020) . The WAVI inversion method yields spatially varying values of two free parameters relating to basal friction and ice viscosity. Since both ice velocities and mass change rates are targeted simultaneously through the cost function, it is likely that there is a trade-off between errors in these quantities and that the modelled mass change rates do not agree with

observations in all locations. Neither Arthern et al. (2015) nor Bett et al. (2023) give a quantitative comparison of modelled and observed mass change rates. More recently, Rosier et al. (2024) used the misfit between observed and modelled mass change rates in their data assimilation inversion, similar to Rosier et al. (2021). They state explicitly that it is not possible to obtain a perfect fit for both ice surface velocities and observed mass change rates, since those two datasets are not mutually consistent.

The other initialization approach is the spin-up method (Winkelmann et al., 2011; Pollard and Deconto, 2012; Greve and Blatter, 2016; Quiquet et al., 2018; Lipscomb et al., 2019; Berends et al., 2021; Berends et al., 2022), which consists of a long (typically several thousands of years) run with paleoclimatic or preindustrial forcing during which the ice sheet can freely evolve. Optionally, uncertain parameter values can be tuned to nudge variables like ice thickness and surface velocities toward present-day observed values. This happens in model runtime: at regular intervals, one or more free

parameters are adjusted as necessary to decrease the misfit between the model and observations. The resulting ice sheet is ideally close to observations and in near equilibrium with little drift (i.e., with little mass change). The modeled ice sheet can also be tuned to historical estimates or observations, and then advanced to the present-day using forcing from global climate

models (GCMs), to avoid an unrealistic steady state at the start of a forward run (Reese et al., 2023; Coulon et al., 2024). This has the advantage of obtaining the modelled mass change rates through a physical process (ocean warming or changes in the surface mass balance), but this adds two uncertainties: the ice sheet geometry from before the observational period, along with the forcing over the historical period.

This study describes an addition to the spin-up method to match observed mass change rates at the start of forward simulations without relying on uncertain historical ice sheet geometry and forcing. This method is similar to the mass balance modification mentioned in Hill et al. (2023), but instead of minimizing mass change rates, we aim to match the observed rates. These rates are derived from satellite-observed surface elevation changes (Smith et al., 2020), corrected for firn processes and glacial isostasy. We apply this method to the Community Ice Sheet Model (CISM, Lipscomb et al. (2021); Lipscomb et al. (2019), allowing us to initialize the Antarctic Ice Sheet in its observed disequilibrium. We use this setup to conduct forward simulations without changing the climate forcing. In these simulations, we find that Thwaites Glacier, Pine Island Glacier and eventually large parts of the WAIS collapse over many centuries. We run similar simulations using the Utrecht FinitE voluMe Ice Sheet Model UFEMISM (Berends et al., 2021) to show that the method can be easily implemented across models and to make sure that WAIS collapse is not unique to CISM.

Section 2 describes the two ice sheet models, the initialization method, and the subsequent forward runs. In Sect. 3, we show that the new initialization procedure results in a modeled present-day state in CISM which is in very good agreement with observations (ice thickness RMSE 34 m, ice surface velocity RMSE = 121.2 m/yr, GL position on average within 1.5 km from the observed GL), and importantly has the same disequilibrium as observed by Smith et al. (2020). In Sect. 4, we discuss the default forward run, starting from the initialized state, under sustained present-day atmosphere and (inverted) ocean forcing, purposely excluding further climate forcing. We repeat the default simulation with UFEMISM (Sect. 5) to show that the observed retreat of TG and PIG is consistent in the two models. In Sect. 6, we show that the CISM behavior is robust under a wide variety of physical approximations and parameter settings. The manuscript ends with a discussion in Sect. 7 and a conclusion in Sect. 8.

## 2 Methods

### 2.1 CISM

The Community Ice Sheet Model (CISM; (Lipscomb et al., 2021; Lipscomb et al., 2019) is a thermo-mechanical higher-order ice sheet model, which is part of the Community Earth System Model version 2 (CESM2, Danabasoglu et al. (2020). This study primarily uses CISM v2.1 and builds on earlier applications of CISM to Antarctic Ice Sheet retreat (Lipscomb et al., 2021; Berdahl et al., 2023). CISM is run at 4 km resolution with a vertically integrated higher-order approximation to the momentum balance (Goldberg, 2011; Robinson et al., 2022), and a regularized Coulomb sliding law (Zoet and Iverson,

2020) as basal boundary condition. We apply a surface mass balance and surface air temperature climatology spanning 1979-2016 from RACMO version 2.3p2 (Van Wessem et al., 2018). Basal melt rates are calculated using a quadratic relation with

a thermal forcing observational dataset (Jourdain et al., 2020). Internal ice temperatures are allowed to evolve freely during the initialization and the forward runs. The calving front is maintained at its present day position. This conservative approach has been chosen because calving physics as well as the impact of ice berg melange of ice shelves are still poorly understood, and overly retreating calving fronts enhance mass loss which make the results overly negative. Variable names and parameter values used in the following text are summarized in Tables 1 and 2.

Table 1. Variables used in this study.

| Variables | Definition |
|---|---|
| $b$ | Bedrock height, positive above sea level |
| $b_{mlt}$ | Basal melt rates under floating ice |
| $C_c$ | Basal friction coefficient (Coulomb C) (inverted quantity in this study) |
| $C_r$ | Basal friction relaxation target |
| $H$ | Modeled ice thickness |
| $H_f$ | Ice thickness above floatation |
| $H_{obs}$ | Observed ice thickness |
| $N$ | Effective pressure |
| $TF_{base}$ | Thermal forcing applied at the ice shelf draft |
| $u_b$ | Basal ice speed |
| $\delta T$ | Ocean temperature correction (inverted quantity in this study) |
| $\tau_b$ | Basal shear stress |

Table 2. Parameters and their units and values used in this study.

| Parameters | Values | Units | Definition |
|---|---|---|---|
| $c_{pw}$ | 3974 | J kg$^{-1}$ K$^{-1}$ | Specific heat of seawater |
| $g$ | 9.81 | m s$^{-2}$ | Gravitational acceleration |
| $H_0$ | 100 | m | Ice thickness scale in the inversion |
| $L_f$ | $3.34 * 10^5$ | J kg$^{-1}$ | Latent heat of fusion |
| $m$ | 3 | - | Basal friction exponent |

| $T_r$ | 0 | K | Relaxation target of the ocean temperature inversion |
|---|---|---|---|
| $u_0$ | 200 | m yr$^{-1}$ | Yield velocity in the basal sliding law |
| $p$ | 0.5 | - | Exponent in effective pressure relation |
| $r$ | 0.05 | - | Strength of inversion relaxation |
| $\rho_i$ | 917 | kg m$^{-3}$ | Density of ice |
| $\rho_w$ | 1027 | kg m$^{-3}$ | Density of ocean water |
| $\tau$ | 100 | yr | Time scale in the friction inversion |
| $\gamma_0$ | 30000 | m yr$^{-1}$ | Basal melt rate coefficient |
| $\tau_o$ | 25 | yr | Time scale in the ocean temperature inversion |

### 2.1.1 Basal friction

In the default configuration, basal friction is parameterized using a basal sliding law based on laboratory experiments (Zoet and Iverson, 2020). This sliding law combines elements of ice sliding on a hard bed (e.g., bedrock) with sliding on deformable soft beds (e.g., saturated till). Its end cases are Weertman-style sliding, with basal velocities dependent on some power of the basal friction, and Coulomb sliding, with basal friction and velocities decoupled. The former mechanism usually dominates where the ice sheet is flowing on hard, non-deformable beds, while the latter describes fast-flowing outlet glaciers with softer deformable till such as Pine Island and Thwaites Glaciers. In CISM, the basal sliding law is implemented as follows:

$$\tau_b = C_c N \left( \frac{u_b}{u_b + u_0} \right)^{\frac{1}{m}} \tag{1.1}$$

where $C_c$ is a unitless parameter controlling the strength of the Coulomb sliding; In CISM, $C_c$ corresponds to $tan\,\phi$ of Zoet and Iverson (2020), in which $\phi$ is the friction angle, a material property of the subglacial till. Since $C_c$ is poorly constrained by theory and observations, we use it as a spatially variable tuning parameter, which we tune in the following way using a nudging method (Lipscomb et al., 2021):

$$\frac{dC_c}{dt} = -C_c \left[ \left( \frac{H - H_{obs}}{H_0 \tau} \right) + \frac{2}{H_0} \frac{dH}{dt} + \frac{r}{\tau} ln \frac{C_c}{C_r} \right] \tag{1.2}$$

in which $H$ is the modeled ice thickness, $H_{obs}$ the observed ice thickness, and $C_r$ the relaxation target for $C_c$. $H_0$, $\tau$ and $r$ are empirical constants used to vary the relative magnitude of each term. The third term in brackets, which is not included in Lipscomb et al. (2021), provides a relaxation target $C_r$ that helps prevent $C_c$ from drifting toward extreme values (limited to $10^{-6}$ and 1). It is a function of elevation, with lower values at low elevation where soft marine sediments are likely more

prevalent. We chose targets of 0.1 for bedrock below -700 m asl and 0.4 for above 700 m asl, with a linear interpolation in between, similar to Aschwanden et al. (2013). Our values for $H_0$ and $\tau$ are respectively 100 m and 100 yr.

The effective pressure $N$ in Eq. (1.1) is the pressure at the ice–bed interface, equal to the difference between the ice overburden pressure and the subglacial water pressure. The effective pressure is lowered near GLs to represent the connection of the subglacial hydrological system to the ocean (Leguy et al., 2014):

$$N = \rho_i g H \left(1 - \frac{H_f}{H}\right)^p \tag{1.3}$$

where $\rho_i$ is the density of ice, $g$ is gravitational acceleration, $H$ is the ice thickness, and $p$ is an exponent between 0 and 1, with a larger value implying a stronger ocean connection. A value of p=0.5 is chosen in this study unless stated otherwise, to include some hydrological connection (Leguy et al., 2021; Lipscomb et al., 2021). Different values are tested in the sensitivity analysis. The flotation thickness $H_f$ is the height of an ice column resting on bedrock below sea level ($b < 0$) at hydrostatic equilibrium:

$$H_f = \max\left(0, -\frac{\rho_w}{\rho_i} b\right) \tag{1.4}$$

where $\rho_w$ is the density of sea water and $b$ is the bedrock elevation.

**2.1.2 Basal melt rates**

Beneath grounded ice, CISM computes basal melt rates based on the net of geothermal, frictional, and conductive heat fluxes at the bed. In our runs, however, there is no coupling between basal water and sliding parameters. Beneath floating ice, we

use the ISMIP6 local parameterization (Jourdain et al., 2020):

$$b_{mlt} = \gamma_0 \left(\frac{\rho_w c_{pw}}{\rho_i L_f}\right)^2 \left(\max[TF_{\text{base}} + \delta T, 0]\right)^2 \tag{1.5}$$

The max function ensures non-negative melt rates; negative rates imply refreezing (accretion), which is not included. We tune $\delta T$ to optimize the ice thickness agreement with observations, similar to Eq. (1.2):

$$\frac{d(\delta T)}{dt} = -\delta T \left[\left(\frac{H - H_{obs}}{H_0 \tau_o}\right) + \frac{2}{H_0} \frac{dH}{dt}\right] + \frac{(T_r - \delta T)}{\tau_o} \tag{1.6}$$

where $T_r$ is a relaxation target for the temperature correction. Ideally, the correction is small, and therefore we set $T_r = 0$ everywhere. Melt rates are sensitive to $\gamma_0$ in Eq. (1.5). We chose a default value $\gamma_0 = 3.0 \times 10^4$ m/yr, for which the average $\delta T$ in the TG and PIG basins at the end of the transient spin-up is close to 0. In other words, we assume that the observed ocean

temperatures in this region are of the right magnitude to drive basal melt rates consistent with observations. Our value for $\gamma_0$ lies about halfway between the median "MeanAnt" and "PIGL" calibration values suggested by Jourdain et al. (2020). The depth dependency is introduced in the dataset $TF_{\text{base}}$ from Jourdain et al. (2020) in two ways. First, the thermal forcing increases with depth due to the decrease of the pressure melting point. Second, in the dataset of Jourdain et al. (2020), there is warm CDW present in the ASE leading to increased thermal forcing with depth. In a forward run, we keep $\delta T$ constant at the value computed during the spin-up. When the grounding line retreats, we extrapolate basin-average values of $\delta T$ (basins are defined following Zwally et al. (2015)), to newly floating cells.

### 2.1.3 Grounding line subgrid-scale parameterization

The GL is not explicitly modeled in CISM but follows from the hydrostatic balance. The modeled GL cuts through cells, with some cells partly grounded and partly floating. To prevent abrupt jumps in modeled quantities at the GL, we use a GL parameterization (Leguy et al., 2021). We define a flotation function,

$$f_{\text{float}} = -b - \frac{\rho_i}{\rho_w} H \tag{1.7}$$

which varies smoothly from negative for grounded ice to positive for floating ice. (For floating ice, $f_{\text{float}}$ is the water column thickness in the cavity.) $f_{\text{float}}$ is used to compute the floating fraction as a percentage of grid cell area by bilinearly interpolating its value from cell vertices to the cell areas scaled to the cavity thickness (Leguy et al. (2021). The grounded and floating fraction of a cell are then used to scale basal friction and basal melting. For the basal melting, this is referred to as the Partial Melt Parameterization (PMP). Other parameterizations tested in the sensitivity experiments of this study are the Floatation Condition Melt Parameterization (FCMP), in which full melt is assigned to a cell once the cell center is floating, and the No Melt Parameterization (NMP), in which no melting is allowed in any cell that is partly grounded.

### 2.1.4 Calving

There are several calving laws in the literature (e.g., (Yu et al., 2019); Wilner et al. (2023); Greene et al. (2022). However, there is no agreed-upon best approach to Antarctic calving (this holds for the Greenland Ice Sheet as well; see for example Benn et al. (2017)), and most calving laws struggle to reproduce the observed calving front at multiple locations simultaneously without adjusting local parameters (Amaral et al., 2020). We therefore choose to apply a no-advance calving scheme, preventing the calving front from advancing beyond the observed present-day location. The calving front can retreat, but only when the ice thins below a threshold thickness of 1 m. In most places, it does not move. This is a conservative approach for a retreating system; in practice, we would expect the calving front to move inland and therefore provide less buttressing for upstream ice. By taking a conservative approach, we rule out the possibility that a simplistic calving algorithm is (partly) responsible for the modelled retreat.

## 2.2 Initialization methods

Typically, CISM nudges basal friction parameters and ocean temperatures to optimize the agreement of ice thickness with observations, allowing a modest misfit (a misfit scale of $H_0 = 100$ m in Eqs. (1.2) and (1.6)) to avoid overfitting, but without considering the observed mass change rates. We refer to this method as the 'equilibrium spin-up. This results in a steady state, where the integrated SMB is in equilibrium with the integrated ice losses: the modeled calving and basal melt fluxes. There is no isostatic adjustment in the default experiments; we assume a static bedrock. The model converges within ~$10^4$ model years, which is typically assessed by considering several criteria like change in ice mass ice area and ice velocities. A converged model state should not drift significantly when run forward with fixed inverted quantities. We calculate the model drift in the modelled ice sheet after stopping the inversion (i.e. by continuing the simulation with the inverted fields kept constant) for 2000 years. We accept an initialization as 'stable' once there is little to no instantaneous model drift and there is no TG and PIG collapse after 2000 years. Then, it is certain that a PIG and TG collapse is caused by applying present-day mass change rates, and not influenced by model drift from the initialization.

The addition to the equilibrium spin-up, referred to as the transient spin-up, allows the model to reach a geometry in agreement with observations, similar to the result of the equilibrium initialization, while also allowing the ice to thin or thicken in agreement with satellite observations at the start of the forward run (i.e., at the present day). Our goal is to create the observed dynamic disequilibrium, so we want to include the observed mass change rates as a target. To hold the ice geometry near observed values during the spin-up, we apply a correction term equal to the negative of the observed mass change rates. That is, we artificially apply a thickening term to ice that is thinning and a thinning term to ice that is thickening. This procedure trains the model to produce mass change rates that are (nearly) identical to the observations during the subsequent forward run, when the correction term is removed.

The mass balances for respectively the equilibrium initialization and the transient spin-up, respectively, are given by

$$\frac{\delta H}{\delta t} = -\nabla F + B \tag{1.8}$$

$$\frac{\delta H}{\delta t} = -\nabla F + B + C \tag{1.9}$$

In these equations, $\nabla F$ is the ice flux divergence, $B$ the sum of the basal and surface mass balances, and $C$ the correction term applied to hold the ice at the observed thickness during the spin-up. Ideally, the initialization procedure ends when the left-hand sides of Eqs. (1.8) and (1.9), $\frac{\delta H}{\delta t}$, are zero. Hence, at the end of the equilibrium initialization, which uses Eq. (1.8), $\nabla F$ is being balanced by $MB$, and forward simulations start with $\frac{\delta H}{\delta t} \approx 0$. In Eq. (1.9), used for transient initialization, we end the

initialization when $\frac{\delta H}{\delta t} \approx 0$. When we remove the $C$ term, forward simulations start with $\frac{\delta H}{\delta t} \approx -C$, where the right-hand side is equal to the observed mass change rate, as desired.

The inversion methods are stopped when the modelled ice sheet has little to no model drift once run forward with the inverted friction and ocean temperatures, and particularly no significant ice mass loss in the ASE basin. The modelled ice sheet should ideally have little instantaneous drift compared to the observed mass change rates, and no thinning or collapsing glaciers during a continuation run of 2000 years with fixed inverted quantities and no mass change rates applied.

During a 2000 years model drift experiment, the model tends to grow to a slightly advanced state for PIG and the Dotson ice shelf, see Fig S1, S2, and S3. This advance is mostly due to the local positive feedback between the ice shelf geometry and basal melt rates that amplifies any small residual drift. Model drift is largely reduced in a forward run with a time-constant instead of geometry based (e.g. Eq. 1.5) basal melt rate field (not shown). Initialization takes about $10^4$ model years.

## 2.3 Forward simulations

Forward simulations are continued without additional climate forcing, to assess the impact of the present-day ocean thermal forcing on the evolution of the AIS, and thereby the sea-level contribution over time. The future simulation using the transient spin-up method is referred to from now on as the "default experiment". As the thermal forcing at a given ocean depth does not
change, the modeled future evolution of the ice sheet is the response to the present-day forcing.

In forward simulations, the ice sheet starts to retreat after the mass correction term in the spin-up is removed. This has implications for the inverted $C_c$ and ocean temperatures in cells that change from grounded to floating or vice versa. If grid points initialized as grounded ice become afloat, we extrapolate the basin-average ocean temperature corrections under the
260 shelf from the end of the initialization. If initially floating grid points become grounded, we use an elevation-dependent parameterization to estimate the local $C_c$, similar to what was done in Aschwanden et al. (2013) and the same as what was used to calculate the relaxation target $C_r$ .

The default forward simulation is compared against an extensive set of sensitivity experiments with modified physics and
265 parameter choices. We designed these experiments conservatively, with many model choices that would tend to stabilize the modelled ice sheet and could potentially stop PIG and TG from collapsing. Section 6 and the supplementary material describe the sensitivity experiments.

## 2.4 UFEMISM

We repeated the default experiment with the transient spin-up using the Utrecht FinitE VoluMe Ice Sheet Model (UFEMISM) (Berends et al., 2021). We use the same transient initialization procedure including the mass change rates from Smith et al. (2020) in the mass balance equation, and climate forcing as in the default CISM experiment, but apply a different momentum balance approximation, basal friction law, calving algorithm, and discretization scheme.

UFEMISM is a thermo-mechanical ice-sheet model designed for long-term simulations at variable resolution, e.g. simultaneously high resolution at the grounding line and lower resolution in the slow-moving interior (Berends et al., 2021). In this study, UFEMISM discretizes the hybrid SIA/SSA approximation to the stress balance (Bueler and Brown, 2009) and other physical equations on a dynamic adaptive triangular mesh. Mesh resolution ranges from ~4 km at the grounding lines of PIG and TG, to ~60 km for slow-moving inland ice. We apply a Budd-type sliding law, see Eq. (1.10), which contains spatially-varying friction coefficients that aim to represent subglacial substrate conditions:

$$\tau_b = tan(\phi)\, u_b^{\frac{1}{m}} N^q \qquad\qquad (1.10)$$

We invert for $\phi$ by nudging toward a target geometry (Bernales et al., 2017b; Pollard and Deconto, 2012). As part of this nudging method, UFEMISM simultaneously inverts for sub-shelf melt rates and the corresponding ocean temperatures needed to reproduce the target geometry over ice-shelf sectors (Bernales et al., 2017a). Both calibrations are performed during model initialization, which brings the modeled ice sheet to a state of equilibrium with a prescribed set of boundary conditions. For this study, friction coefficients and ocean temperatures are inverted during the first 90,000 model years of the initialization, after which their values are kept fixed in time. The model is then run for another 10,000 years, which allows it to evolve unperturbed until equilibrium is attained. From 100,000 years, the default experiment with the mass change rates is started. We use both the PMP (default) and FCMP subgrid-scale melting parameterizations from Leguy et al. (2021), as described above. The calving front is conservatively fixed at its present-day position,

## 3. Results: Transient present-day state

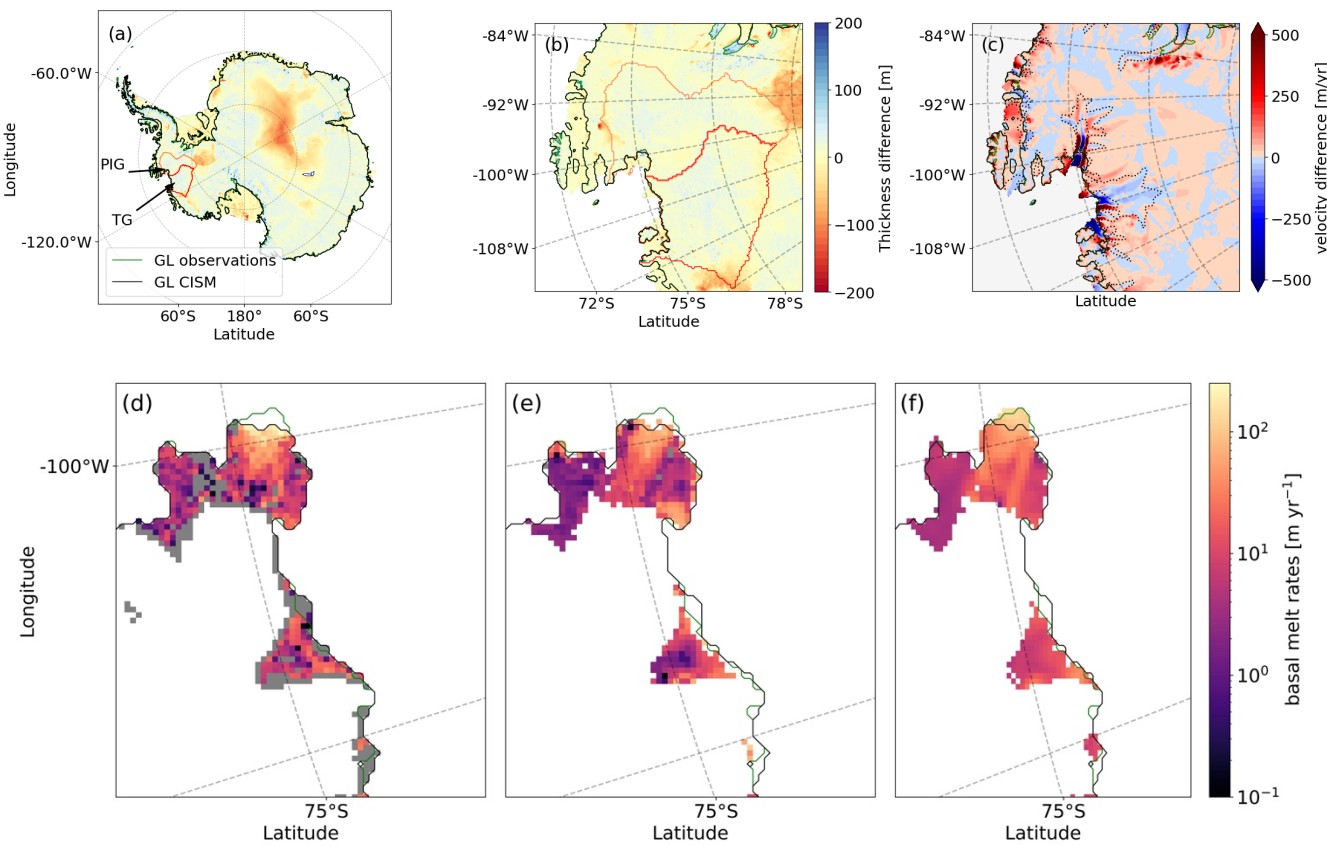

**Figure 1. Initialized state of the Antarctic Ice Sheet using CISM.** (Top row) Ice thickness difference between the final state of the transient
initialization and observations for all of Antarctica (RMSE = 34 m, (a)) and the ASE region (RMSE = 19.3 m, (b)). The black line represents
the modeled GL. The observed GL (green) is visible only where the modeled and observed GL deviate in position, for example at the PIG
GL. Red and orange contours are drawn around the TG and PIG basins, respectively. (c) Surface ice velocity difference with respect to
observations of Rignot et al. (2019) (RMSE = 156 m/yr). Positive differences indicate locations where CISM overestimates velocities.
Observed surface velocities of 100 (outer) and 500 (inner) m/yr are contoured by black dots, to highlight fast-flowing regions with large
velocities and gradients. For the ASE, the velocity RMSE = 99 m/yr. For ASE shelves only, the velocity RMSE = 216 m/yr. (Bottom row
panel d,e,f) Basal melt rates, with grey and white denoting zero basal melt rates and no data, respectively, of (d) CISM with an integrated
flux of 106 Gt/yr), (e) Adusumilli et al. (2020) with an integrated flux of 94 Gt/yr, and (f) Rignot et al. (2013) with an integrated flux of 149
Gt/yr.

Figure 1 shows the modeled CISM state at the end of the transient initialization. The thickness error over the ASE region is
low (RMSE = 19.3 m), and the GL closely follows observations, with an average 1.5 km difference (calculated as the average

distance between the modeled GL position and the closest observed GL position). The modeled GL for PIG is shifted seaward by 5–10 km. The modeled basal melt rates and melt patterns beneath floating ice agree well with the values from Rignot et al. (2013) and Adusumilli et al. (2020), with an integrated melt flux within the range of the two datasets. The instantaneous and average model drift of this simulation is shown in Fig S1, S2 and the inverted quantities of both initializations are shown in S3. The initialized state in UFEMISM is shown in Fig. S4 and the inverted quantities in S5, and the result of the CISM equilibrium initialization in Fig. S6.

The modeled ice surface velocities are in good agreement with observations, even though they are not a target for assimilation. The RMSE of the ice surface velocities is comparable to other ISMIP6 models (Seroussi et al., 2020), of which the range is 100 – 400 m/yr. Many of those models are optimized to match observed velocities and use a variety of spin-up and data assimilation methods. Modeled PIG and Thwaites velocities are too low midstream and too high in the lateral margins, possibly because of the lack of brittle processes in CISM. These regions are heavily crevassed (Lhermitte et al., 2020); taking crevasses into account would weaken the ice, speed up the main flowlines, and slow down the margins, because the margins are then decoupled from the center flowline. The center flowline can accelerate and is not necessarily 'dragging along' the slower moving margins, which would bring our modelled ice surface velocities closer to observations. The integrated ice fluxes across the GLs of TG and PIG are close to the observations of Morlighem et al. (2020) (Fig. S7). As the integrated ice flux is in good agreement with observations, we believe that the dynamical characteristics of the model with a very accurate GL position and ice thickness and ice thickness changes almost equal to observations, are sufficient for our continuation experiments.

In the observations of Smith et al. (2020), the ASE shelves are thinning rapidly. Simulating these large negative mass change rates here requires higher ocean temperatures and lower basal friction under grounded ice compared to the equilibrium initialization. The greater friction during the equilibrium initialization leads to an underestimation of ice velocities (Fig S6). As a result, the grounding line flux is lower (see Fig. S7); therefore, lower basal melt rates and in general colder ocean conditions are required to match the observed ice shelf geometry.

The observed and modeled ice thickness change rates in the first years after the start of a default continuation run are shown in Fig. 2. The modelled mass change rate pattern over the first year (b) mimics the observations (a) in detail. After 5 years, the thinning rate of PIG decreases slightly, while the thinning rate of TG increases slightly. The overall pattern of the modelled rates changes little during the first timesteps of a continuation simulation, highlighting that CISM is able to reproduce the observed rates robustly.

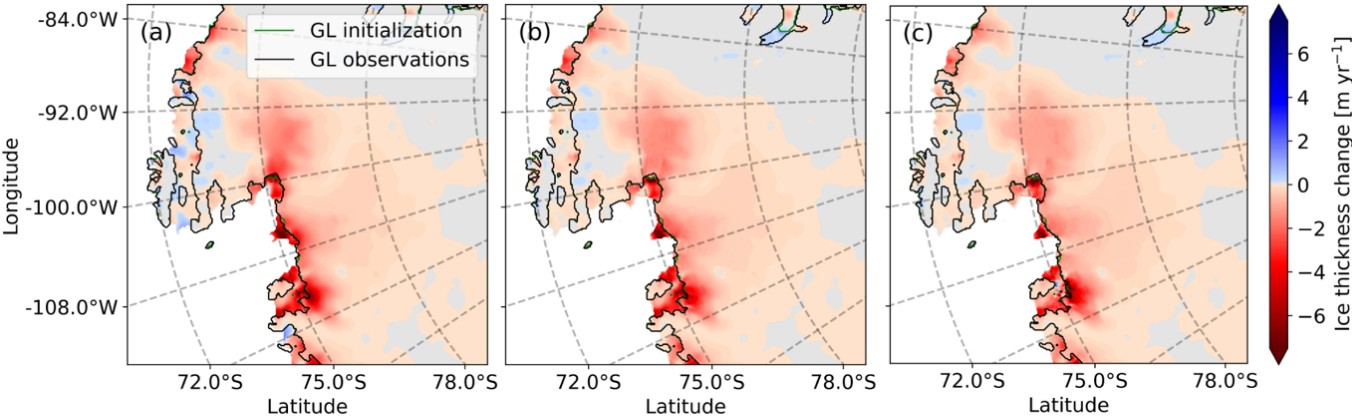

**Figure 2. Observed and modelled present-day mass change rates** (a) Ice thickness change rates in the observations of Smith et al. (2020) with the observed grounding line in green and the modelled grounding line at the end of the transient initialization in green, in the ASE region. Modelled ice thickness change rates when starting a continuation run from the transient initialization after 1 yr (b), and 5 yr (c).

## 4. Results: Future state

In the default experiment with present-day climate forcing, AIS mass loss is dominated by the collapse of TG (including the neighboring Smith, Pope, and Kohler glaciers) and PIG (Fig. 3). In the first four centuries, mass loss is comparable to the current observed rates (Fig. 3b). During this period, the total AIS mass loss averages 0.5 mm GMSL yr$^{-1}$, slightly less than the current combined loss from PIG and TG, because there is currently some East Antarctic Ice Sheet thickening, primarily in Dronning Maud Land (Smith et al., 2020), which continues in the forward simulation (Fig. 3a). Areas where little to no mass changes are observed in Smith et al. (2020), such as the Siple Coast and Victoria Land, remain in balance in our simulation.

After 500 years, the mass loss accelerates as both PIG and TG collapse. We can identify a shallow ridge in the TG bedrock profile approximately 45 km upstream of the present-day GL, which acts as the last pinning point before this accelerated collapse begins (see line *AB* in Fig. 3a and a zoom-in in Fig 4.). As soon as the grounding line passes over the line *AB* in Fig 3a and Fig 4, the high ridge approximately 45 km upstream of the present-day grounding line, the collapse starts. This second ridge at about 100 km upstream in the cross section *CD* of similar height as the ridge *AB,* but less extended in the cross-flow direction and surrounded by troughs, see Fig 4.

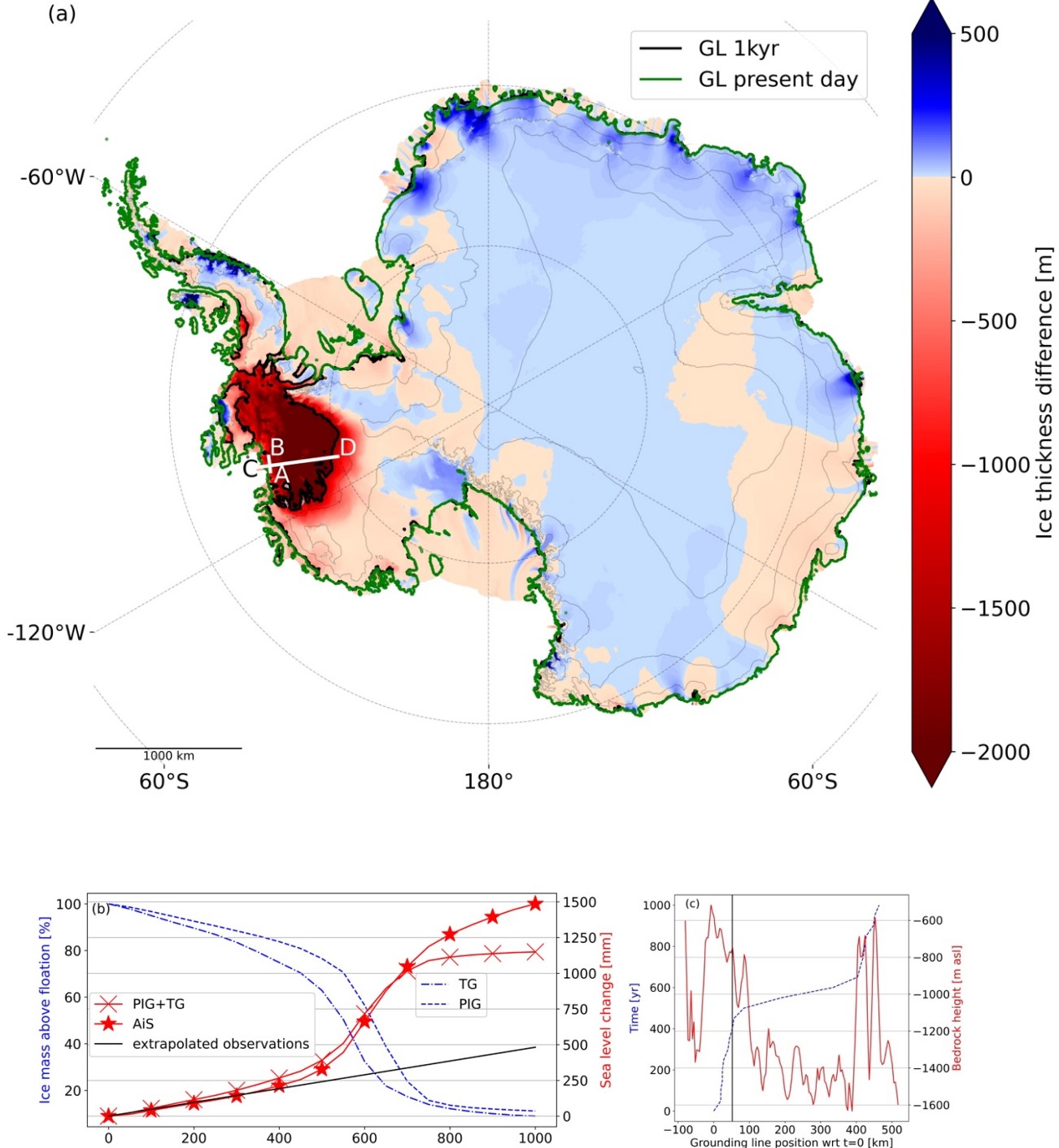

**Figure 3. Future state of the Antarctic Ice Sheet using CISM.** (a) Ice thickness change after a 1000-year continuation experiment initialized with present-day ocean thermal forcing. The original modeled GL resulting from the spin-up is shown in green, and the GL

position at $t = 1000$ years is shown in black, only where it deviates from the spin-up GL. The thin grey lines show the modeled elevation with an interval of 1000 m at t=1000 years. (b) Ice mass above floatation for Thwaites Glacier (TG) and Pine Island Glacier (PIG) in blue, relative to the initial state shown in Fig. 1, and the corresponding sea level change equivalent (calculated according to Eq 2 in Goelzer et al. (2020) of TG, PIG (combined, crosses) and the AIS (stars) in red. Present-day observed mass loss is extrapolated in time and labeled as extrapolated observations in black. (c) GL displacement over time, and the bedrock profile along cross section *CD*. The last ridge before collapse is initiated, *AB*, is shown with a vertical black line. Once the GL passes this ridge, mass loss accelerates. A zoomed-in figure including the 2D bedrock profile is shown in Extended Data Fig. S8.

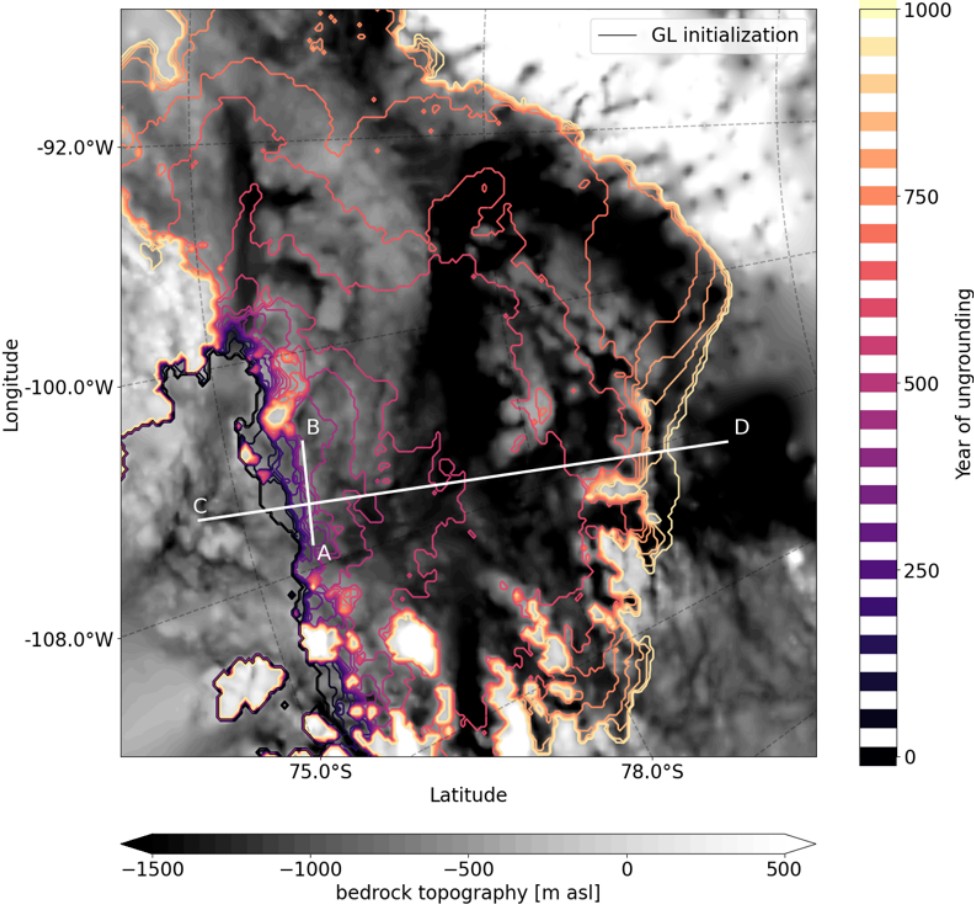

**Figure 4. ASE bedrock topography and grounding line migration of the future simulation.** The bedrock elevation (m) is shown with grey shading. Magma contours show the grounding line position at different times during the default continuation experiment. Contour lines clumped together highlight areas where the grounding line stabilizes for long periods. The short white line (AB) shows the last prominent ridge before the accelerated collapse starts, and the long white semi-horizontal white line (CD) shows the location of the cross section plotted in Fig. 3.

The TG GL recedes faster than that of PIG (Fig. 5b), in line with the mass loss above flotation (Fig 3b), where the TG mass loss acceleration precedes the PIG acceleration. In model year 500, ice velocities in the main channel of TG exceed 4000

m/yr, about twice the current modeled values (Fig. 5e). Thinning is greatest near the new GL, which is on a retrograde bed and therefore deeper than the current GL. Ocean thermal forcing in the ASE increases with depth, therefore newly ungrounded grid cells have a higher basal melt rate and thin rapidly. After 750 years, TG and PIG have collapsed, and the basins contain almost no grounded ice, except on a few small ridges with bedrock above sea level. A large, confined ice shelf has formed, with a thickness of several hundred meters near the GLs, decreasing to tens of meters near the (non-moving) calving front. At

that time, the two basins have contributed about 1.2 meters to global mean sea level. Hence, the retreat of TG and PIG consists of two phases: a gradual decrease for the first few centuries, followed by an accelerated collapse, which continues until the basins are empty. PIG retreat is accelerated by the simultaneous collapse of TG. However, each glacier collapses on its own when the observed mass change rates are applied only to a single basin (not shown).


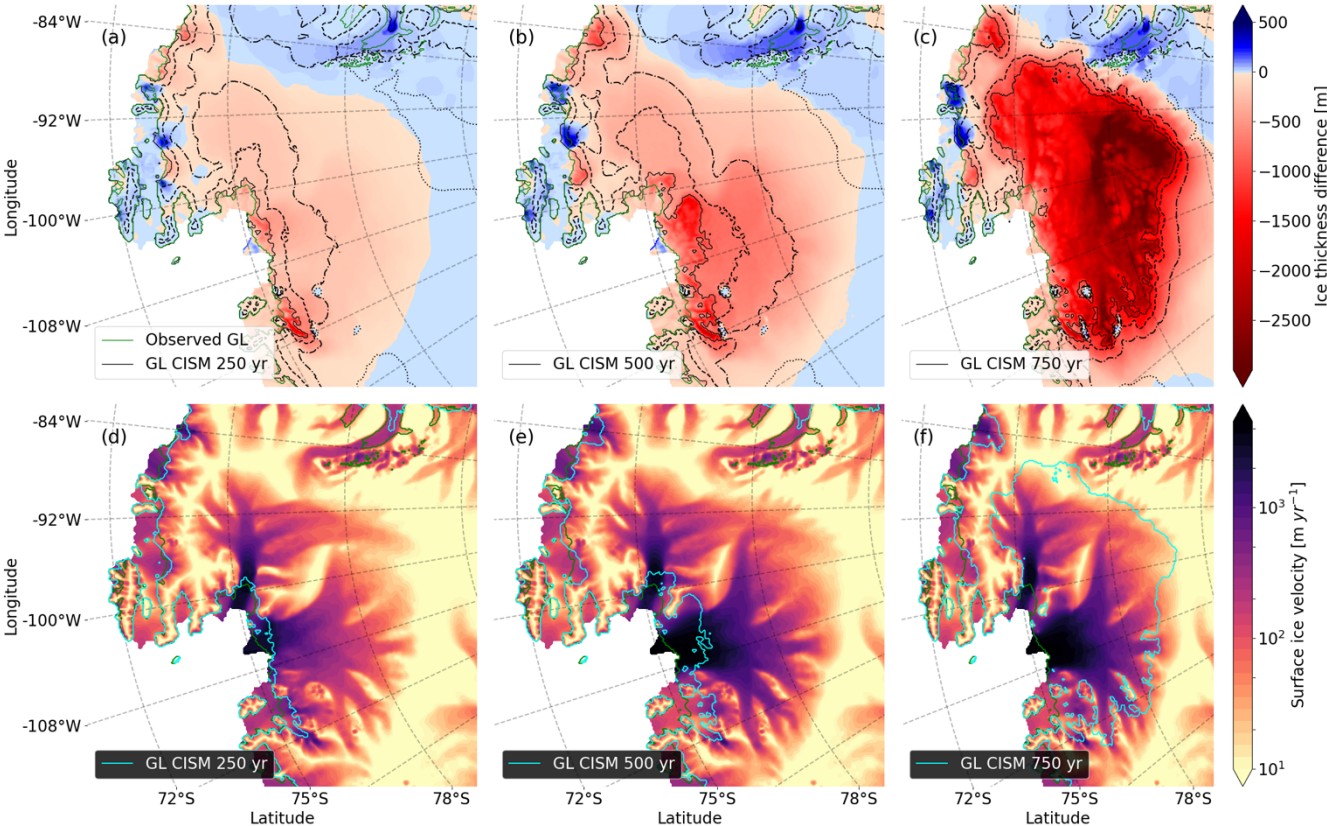

**Figure 5. Development of the collapse at three times.** (Top) Ice thickness difference with respect to the end of the initialization at t=250 yr (a), t=500 yr (b) and t=750 yr (c) after the start of the default continuation experiment. The present-
day GL from observations is shown in green, and the modeled GL is shown in black. The ice surface elevation lines are contours of 500 (dashed), 1000 (dash-dotted) and 2000 (dotted) meters. (Bottom) Ice surface velocities at t=250 (d), t=500 (e) and t=750 (f) years, with the modeled GL shown in cyan, and the observed GL in green.

During the continuation experiment, the collapse can be halted by turning off basal melting completely (yellow lines in Fig.
6a). We observe two features in these zero basal melting experiments: the ice sheet re-advances more slowly than it collapsed, and the re-advance is slower when basal melt rates are switched off later during the simulation. Setting the basal melt rates to zero is equivalent to decreasing ocean temperatures up to 4 K near the GL and on average 2–3 K elsewhere. We also tested the effect of an instant cool-down of 1, 1.5 and 2 K at 250 years (brown lines) and 1.5, 2, 2.5 and 3K after 500 years (red lines) during the simulation. None of the cool-down experiments were enough to regrow the ice sheet to the initial condition on the
same timescales as the retreat. Most of the experiments merely postpone the total collapse of the TG and PIG basins. Only a full basal melt stop will at any moment halt the collapse. We also tested a percentage decrease in basal melt rates at certain timesteps. Setting the basal melt rates to 75% and 50% of their present-day value (yellow and brown lines in Fig 6b) does not

stop the collapse, but only delays it. Setting melt rates of 25% (red lines) of their present-day value results in eventual regrowth, but only when applied in the first 200 years of the simulation (red lines in Fig 6b).

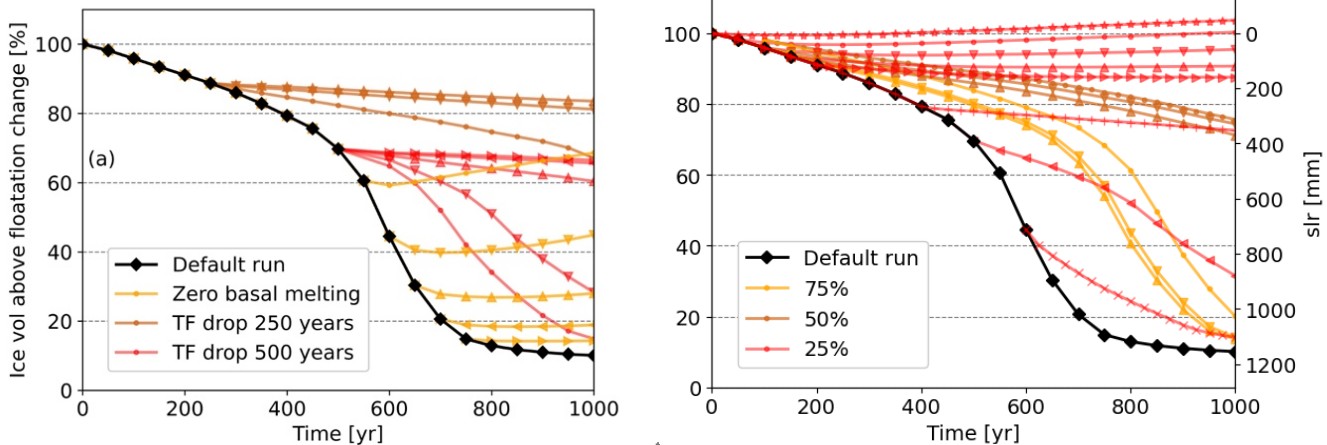


**Figure 6. The impact of decreasing basal melt rates on the ice retreat.** (Left) Evolution of ice volume (shown as a percentage relative to the initial volume (see Fig. 1) for PIG and TG combined and as sea-level equivalent on the right y-axis) over 1000 years in experiments with a sudden decrease in basal melt rates. Brown lines show where the thermal forcing (TF) is instantly decreased after 250 years by 1 K, 1.5 K

and 2 K, and the red lines show simulations where the thermal forcing is decreased after 500 years by 1.5 K, 2 K, 2.5 K and 3 K. Yellow lines indicate runs where the basal melting is instantly switched off after 500, 600, 700 and 800 years. (Right) Percentage of the basal melt rates applied: 25% (red), 50% (brown) and 75% (yellow). The 50% and 75% basal melt rate experiments are started at 50, 100 and 150 years, and the 25% basal melt rate experiments are started at 0, 50, 100, 150, 200, 400, 500 and 600 years after the start of a transient continuation run.


The accelerated collapse resembles the MISI mechanism (Schoof, 2012, 2007). However, the analytical solution of Schoof (2012) and Schoof (2007) does not account for an initial buttressing shelf and is valid only when there is no change in buttressing. Recent studies have indicated that the current TG ice shelves provide little or no buttressing (Gudmundsson et al., 2023; Gudmundsson et al., 2019; Fürst et al., 2016). During and after the modeled collapse, a large buttressing ice shelf forms.

This is at least partly because of our conservative stationary calving front, in reality the front might move inland from its present-day position. As a result, ongoing basal melting is needed to overcome the increased buttressing and sustain the retreat. Removing the basal melt entirely would halt the collapse (see Fig. 6). This makes our retreat at least partly driven by forcing, and not strictly an unstable ice dynamical system, as would be the case in the theoretical non-buttressed MISI description.

It is still debated whether the presence of warm CDW, about 3–4 degrees K above the melting point (Jacobs et al., 2011b), under the PIG and TG shelves is a result of natural variability (Silvano et al., 2022) or forced by climate change (Holland et al., 2019). One study (Holland et al., 2019) found that present-day WAIS basal melting is partly caused by internal variability

and partly anthropogenically forced. In the case of natural variability, CDW might be (partly) absent in the future, lowering the melt rates. When we first run 50 years into the future with the default initialization and then instantly halve the melt rates,
the collapse is delayed but not halted (Fig 6b).

## 5. Results: UFEMISM simulation

The default experiment shown in Fig. 3 is repeated with the ice sheet model UFEMISM. The initialized state of the UFEMISM simulation is shown in Figure S4, in terms of ice thickness and ice surface velocity differences with present-day observations.
The inverted quantities are shown in Figure S5. Figure 7 shows the result of an unforced continuation simulation with UFEMISM, initialized with the present-day mass loss rates and using PMP at the grounding line. The run setup is very similar to the default experiment using CISM but differs, as mentioned in Sect. 2.4, in the stress balance, discretization, sliding law, and grounding line parameterizations.

The ice thickness change shown in Figure 7, after 1000 years of an unforced forward run, is again dominated by the collapse of Thwaites and Pine Island Glacier, just as in the default CISM simulation shown in Figure 3. The same floating corridor has formed at the Rutford ice stream, connecting the Amundsen Sea and the Filchner-Ronne ice shelf. The Antarctic Peninsula is still connected to the Bellingshausen sector via grounded ice, a feature that is absent in the CISM simulation shown in Figure 2. The UFEMISM simulation predicts slightly less grounding line retreat. On the remaining AIS, the ice thickness change is
largely negative and close to zero (on the order of 1–5 meters of total ice thickness change over 1000 years), with the exception of some small outlet glaciers in Dronning Maud land. The mass change patterns on the Ross and Filchner–Ronne ice shelves match the patterns shown in Figure 3. We therefore conclude that the ASE collapse is likely not restricted to CISM, but rather the result of initializing an ice sheet model with the present-day mass change rates.

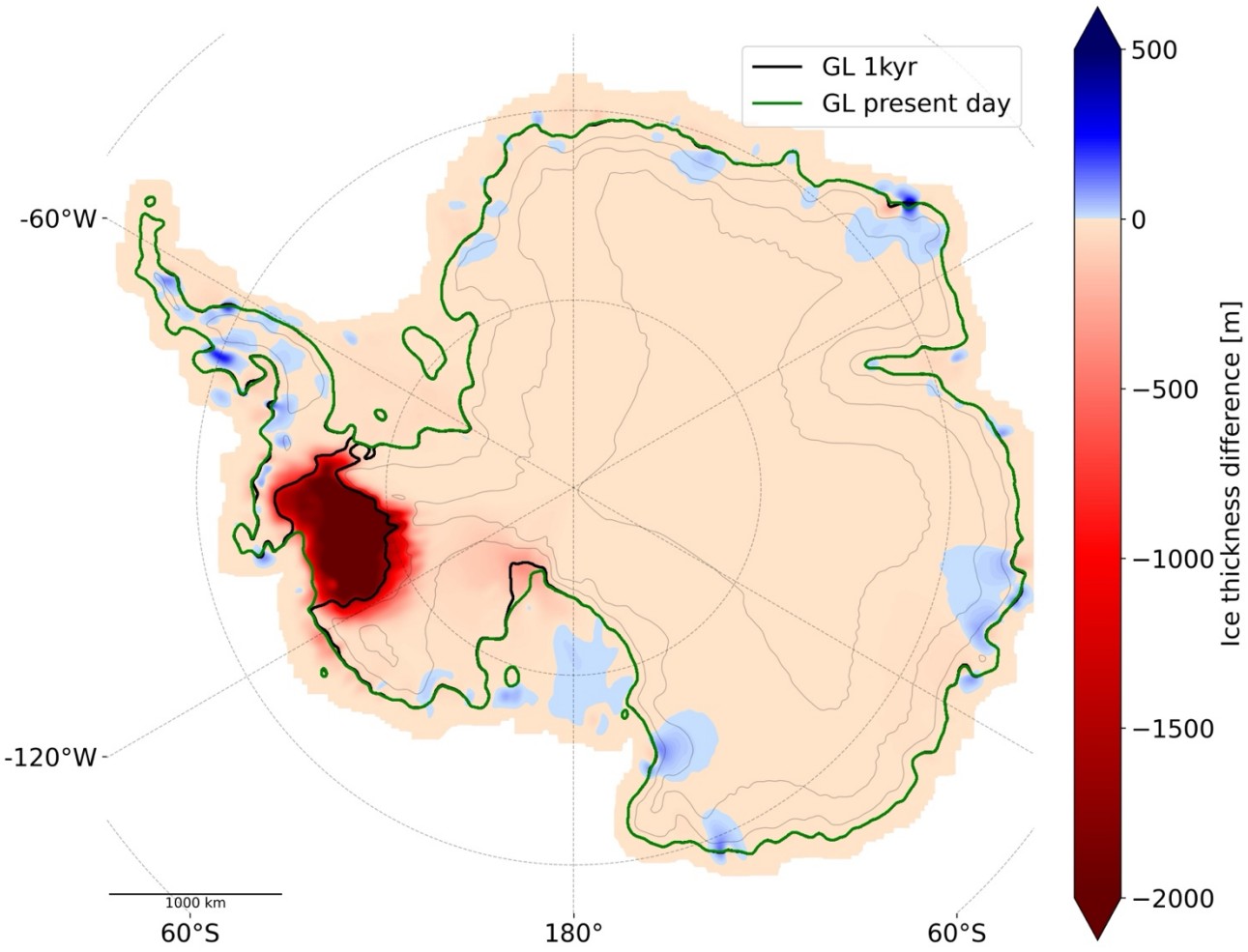


**Figure 7. Future state of the Antarctic Ice Sheet using UFEMISM.** Ice thickness change after a 1000-year continuation experiment initialized with present-day mass change rates. The initial modeled GL resulting from the spin-up is shown in green, and the GL position at *t* = 1000 years is shown in black, only where it deviates from the spin-up GL. For this figure, the UFEMISM data on an irregular triangular

grid were interpolated to the same regular rectangular grid as is used in the CISM simulations, and smoothed with a Gaussian kernel for visibility.

## 6. Model choices and parameter value exploration

The results presented in Section 4 are based on the best estimate of the current system state and a single set of preferred model physics choices and parameter values. To assess the sensitivity of our results to those choices and parameter values, we varied them within ranges we viewed as physically realistic. The aim of these experiments is to determine which processes and parameter choices can either accelerate or slow (and perhaps prevent) the projected collapse. Where appropriate, we redid the transient initialization with the new choices (runs with an "*" in Table 3). In other cases, we

recomputed free parameters to start from the same initialized state (same ice thickness and velocities, and initial dh/dt) as our default experiment, to minimize the effect of a slightly different initialized state. We verified for the simulations starting with a new initialization that there is little model drift, and that PIG and TG would not collapse without applying the mass change rates.

Figure 8 and Table 3 show the relative mass loss and resulting GMSL rise relative to the initial state for this set of model choices and parameter values. For some simulations, marked with an '*' in Table 3, a new initialized stated was necessary. We made sure that this new initial state was close to the default initial state in terms of thickness, velocity and grounding line position RMSE. We ran model drift experiments to verify that for the new initialized state, the WAIS would not collapse without adding the present-day mass change rates. Notably, the continuation of present-day climate forcing leads to TG and

PIG collapse in all experiments, except the default simulation without applying the present-day mass change rates (the model drift experiment, dashed line in Figure 8). In this experiment, PIG starts to readvance to a downstream seabed ridge, creating some positive model drift.

Once a collapse is initiated (i.e., when the mass loss accelerates), mass loss rates are similar, and all experiments show a similar

collapse duration (150–250 years, see Table 3). During the fast retreat phase, the mass loss rate is about 3 mm GMSL/yr for almost every experiment. This is irrespective of the parameter setting, suggesting that the collapse rate during the fast phase is controlled by the bed topography and/or internal system dynamics rather than detailed physics or parameter choices.

The collapse timing is insensitive to the basal friction law (in line with Barnes and Gudmundsson (2022), model resolution (8

km or 4 km), and a 20% increase in the integrated SMB. Also, glacial isostatic adjustment (GIA), estimated using an Elastic Lithosphere Relaxing Asthenosphere (ELRA) model (Lambeck and Nakiboglu, 1980) with a relaxation time of 3000 yr, has little impact on glacier stability. GIA with a very short relaxation time of 100 yr delays the collapse by several centuries but does not prevent it. These findings are similar to those of Berdahl et al. (2023).

The collapse timing is sensitive to model choices related to basal melting. A weaker/stronger dependence of the basal melt on the thermal forcing at the ice draft depth (Jourdain et al., 2020), i.e. a higher/lower value for $\gamma_0$ in Eq (1.5), delays/advances the start of the collapse (as in Berdahl et al. 2023). Allowing no melt in a grid cell containing the GL (NMP) instead of partial melt (PMP) delays the collapse for several centuries, but does not prevent a collapse.

The UFEMISM simulations show a similar pattern of mass loss as the CISM simulations. Again, there is a gradual linear phase of ice mass loss. When approximately 30% of the original ice mass above floatation in the ASE has disappeared, the accelerated collapse begins; this phase lasts approximately 200 years with FCMP and 250 years with NMP. The similar behavior of the UFEMISM simulations suggests it is unlikely that the WAIS collapse modelled in CISM is a model artifact. The timing on the start of the collapse differs in our sensitivity experiment by about 1200 years. The large difference in collapse timing highlights

that the choice of ice sheet model is a major source of uncertainty.

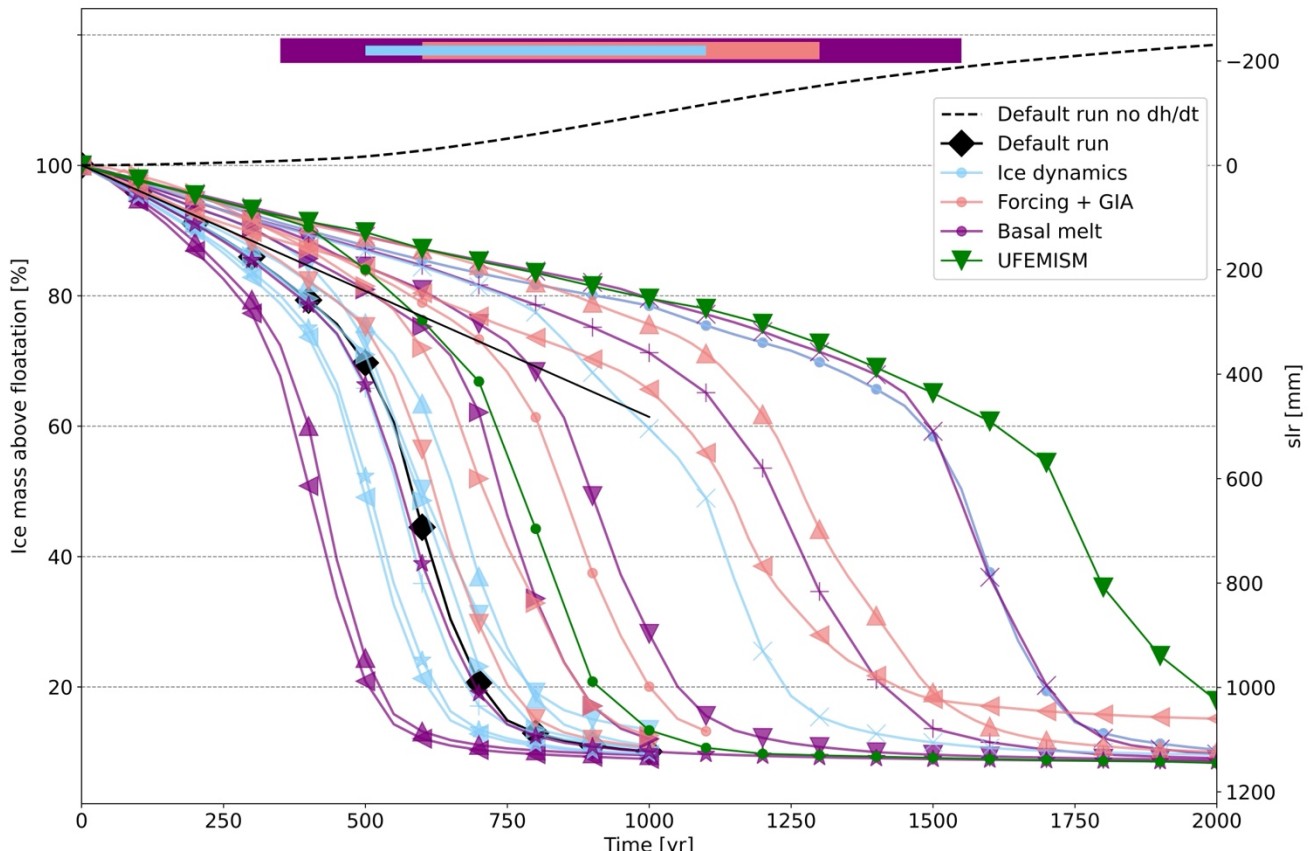

**Figure 8. Sensitivity analysis.** Ice mass above floatation (left y-axis) and equivalent sea level rise (right axis) in the combined Thwaites and Pine Island Glacier basins for different experiments (22 in total). The mass change is calculated as the percentage difference relative to the initial value in the TG and PIG basins. The default run shown in Fig. 3 is plotted with black diamonds. The same run without applying

the present-day mass change rates, i.e. the model drift experiment, is shown as a dashed line. The UFEMISM runs are shown in green, and the extrapolated observations as in Fig. 2b are shown in marker-free black. Most runs start with the observed mass change rates but deviate early in the run due to dynamic feedbacks. Some runs, e.g. "*SMB increases by 20% instantly*" start with different mass change rates. This explains the spreading trajectories in the first 100 years. The colored bars at the top indicate the range of times that the simulations need to reach 50% of their present-day ice volume, to highlight the uncertainty related to different categories. See Table 3 for a description of each run.

**Table 3. Summary of the sensitivity experiments.** Descriptions of the runs shown in Fig. 4. Except as noted otherwise, all settings and resolutions are the same as for the default initialization and continuation run. For some runs, denoted by an "*", a new initialized state was needed. This new initial state is tested on model drift, and verified that a PIG and TG collapse was not found when not applying the present-day mass change rates. An extended description of each run can be found in Extended Data Table S1. The year of collapse initiation is defined as the year when the rate of basin-wide mass loss exceeds a 10% change in initial mass per 75 years (0.13%/year), which is equivalent to 1.72 mm GMSL/yr. The end of the collapse is defined as the first time the mass loss drops below this threshold again.

| Run description (full description in Extended Data Table 1) | collapse initiated [yr] (avg SLR rate [mm/yr]) | collapse completed [yr] (max SLR rate [mm/yr]) | Marker |
|---|---|---|---|
| 8km grid, partial melt parameterization (Leguy et al., 2021) * | 300 (3.45) | 500 (4.41) | ◀ |
| Highest basal melt versus depth sensitivity from Jourdain et al. (2020) | 300 (3.06) | 550 (5.32) | ▲ |
| Larger flow factor (softer ice) * | 400 (3.22) | 600 (4.24) | ◀ |
| No ocean connection (p=0, Eq. 1.3) (Leguy et al., 2021) | 400 (2.74) | 650 (4.24) | ★ |
| With Schoof basal sliding parameterization, based on (Asay-Davis et al., 2016) | 450 (2.66) | 650 (4.33) | + |
| Thermal forcing capped at the maximum found at the GL | 500 (2.69) | 700 (3.9) | ★ |
| Default continuation experiment | 500 (2.95) | 700 (4.15) | ◆ |
| ELRA isostasy, relaxation time = 3000 years | 500 (2.26) | 700 (3.28) | ▶ |
| With a Weertman power-law basal sliding parameterization | 450 (2.51) | 700 (3.01) | ▼ |
| SMB increases by 20% instantly | 500 (2.72) | 750 (3.95) | ▼ |
| Smaller flow factor (stiffer ice) * | 550 (2.54) | 800 (3.82) | ▲ |
| Mass change rates applied only on ice shelves* | 600 (2.16) | 900 (3.18) | ▶ |
| 8-km grid, flotation-condition melt parameterization (Leguy et al., 2021)* | 650 (2.5) | 900 (4.07) | ▶ |
| Decrease basal melt rates after 50 years by 25% | 700 (2.39) | 950 (3.15) | ● |
| UFEMISM, partial melt parameterization* | 700 (2.02) | 900 (3.06) | ● |
| Flotation condition melt parameterization (Leguy et al., 2021)* | 800 (2.36) | 1050 (3.12) | ▼ |
| ELRA isostasy, relaxation time = 100 years | 1100 (2.72) | 1200 (2.2) | ◀ |
| Full ocean connection (p=1, Eq. 1.3) (Leguy et al., 2021) | 1100 (1.82) | 1250 (2.85) | ✳ |
| Lowest basal melt versus thermal forcing sensitivity from ref. (Jourdain et al., 2020) | 1200 (2.72) | 1400 (3.48) | + |
| Decrease basal melt rates after 50 years by 50% | 1200 (2.09) | 1350 (2.59) | ▲ |
| No-melt parameterization at the GL (Leguy et al., 2021) * | 1500 (2.31) | 1700 (3.29) | ● |
| 8-km grid, no-melt parameterization (Leguy et al., 2021) * | 1500 (2.15) | 1700 (2.95) | ✳ |
| UFEMISM, floatation-condition melt parameterization * | 1600 (1.94) | 1850 (2.50) | ▼ |


## 7. Discussion

We have developed a modified ice sheet initialization method and applied it in Antarctic Ice Sheet simulations using two models, CISM and UFEMISM. This method enables an ice sheet model to be spun up to a state consistent with the observed

ice sheet geometry, while also matching the observed mass change rates. Previous initialization methods, both spin-up and data-assimilation method, were unable to accurately represent the observed mass change rates.

In model simulations spanning a wide range of physics and parameter choices, Thwaites Glacier and Pine Island Glacier collapse on a timescale of several centuries without additional climate forcing beyond the ocean forcing required to match
present-day melt rates. In other words, the present ocean-driven mass imbalance is a precursor for large-scale deglaciation of the WAIS. A period of gradual retreat over several centuries is followed by a phase of rapid collapse over about two centuries, with GMSL rising by about 3 mm yr$^{-1}$ during the collapse phase. This behavior, including the collapse rate, is consistent with several previous studies (Joughin et al. (2014); Golledge et al. (2021); Coulon et al. (2024); Feldmann and Levermann (2015)). The timing of collapse initiation differs widely across our experiments, as well as in previous studies, and remains a source of
large uncertainty (Fig 8).

Each sensitivity experiment (Table 3) varies a single parameter or physics choice relative to the default experiment. It would be interesting to study simultaneous changes in multiple parameters and physical choices. However, when combining two conservative model choices, e.g. NMP and low melt sensitivity to thermal forcing, the inversion procedure often fails to
reproduce the observed GL location, suggesting that the combination is unrealistic. For this reason, and because most combinations require a separate, computationally expensive initialization, we did not consider these combined effects. To generate a single conservative end-member, we combined the two most conservative runs ("*No-melt parameterization at the GL*" and "*ELRA isostasy, relaxation time = 100 years*" in Table 1) into a "most conservative estimate." In this simulation, TG and PIG collapse after 2500 years (not shown in Fig. 8). Applying no melt in a cell containing the grounding line is
conservative, since the transition between grounded and floating ice occurs over an extended grounding zone possibly influenced by tides (Graham et al., 2022).

We note three missing processes that could affect the ice evolution. First, the mass change rates on grounded ice are controlled by parameters in the basal sliding law, which are not necessarily constant over time. After the initialization phase, $C_c$ is kept
constant during forward runs, not accounting for future evolution of the ice–bed interface, such as increased basal meltwater that could enhance sliding (Kazmierczak et al., 2024). In addition, possible feedbacks between the ice flow and the bed (e.g. Rémy and Legresy (2004); Van Der Wel et al. (2013)) are only partly represented by the basal friction law and the calculation of the effective pressure on bedrock below sea level, representing the hydrological connection to the ocean.

Second, as ice shelves melt and retreat, they add freshwater to the ocean, but the current model setup ignores the potential feedback of the additional fresh water on ocean temperature or salinity. There are several ways for the freshwater flux from a collapsing WAIS to influence the ocean properties of the Amundsen Sea and the Southern Ocean. For example, increased freshwater can cool the Southern Ocean sea surface (Bintanja et al., 2015) and reduce Antarctic Bottom Water formation

(Williams et al., 2016). Alternatively, the freshwater input can stratify the ocean just beyond the shelves, thus trapping heat in the subsurface ocean and ultimately increasing heat transport into cavities, resulting in higher basal melt rates (Flexas et al., 2022). It has also been suggested that interactions between freshwater flux and sea ice formation and the presence of warm, dense water in large AIS cavities can stabilize and stratify the ocean flow in cavities, preventing further warm water from flowing into the cavity (Hellmer, 2004). Hence, the exact effect of a large freshwater pulse into the ocean surrounding the AIS remains unknown based on model studies (Swart et al. (2023). Since we use a standalone-ice sheet model in this study, and no consensus has been reached on the effect of the freshwater flux, we did not parameterize this effect. In future studies, freshwater feedbacks could be added by coupling the ice sheet model to a cavity-resolving ocean model.

Third, bedrock uplifting could slow down glacier retreat in MISI-prone areas (Book et al., 2022; Kachuck et al., 2020; Van Calcar et al., 2023). In our default experiment, we do not simulate bedrock height changes. We perform sensitivity tests with CISM's ELRA model of glacial isostasy in Section 6. However, these experiments assume, perhaps unrealistically, that the bedrock is in equilibrium at the start of the forward simulation.

## 8. Conclusions and outlook

We have developed a method to incorporate present-day mass change rates in an Antarctic Ice Sheet model initialization using a spin-up. Given these observed rates, we show that for a wide range of model choices, Thwaites Glacier and Pine Island
Glacier will ultimately collapse, likely within 1000 years under present-day conditions. These results are consistent with several previous studies but carried out with an improved initialization scheme, a much wider range of model configurations, and without additional climate forcing. The timing of collapse onset is highly dependent on model details and varies in our study from 300 to 2500 years. The length of the fast collapse phase (about 200 years) and the maximum mass loss during the collapse ($\sim 3$ mm SLE $yr^{-1}$), are consistent across experiments. These experiments suggest that eventual collapse is inevitable
for the current climate and can probably be averted only by a large decrease in sub-shelf ocean temperatures. The collapse will likely begin sooner if Amundsen sub-shelf cavities were to warm further in the next few decades, as projected by a recent ocean modelling study (Naughten et al., 2023).

The present-day observed imbalance in the ASE is likely caused by the presence of warm ocean conditions, and the current
ice sheet geometry is reacting to higher basal melt rates and the loss of buttressing. While some of the warming may be the result of natural variability, it is plausible that the cavities warmed during the past century when a different geometry was present. In that case, GCMs and high-resolution regional models could be used to simulate the coupled ice–ocean–atmosphere processes that might have triggered the recent warming. These models could also explore whether future anthropogenic action could cool these cavities enough to prevent collapse. Replication of our experiments with other ice sheet models could
substantiate whether the collapse of Thwaites and Pine Island Glaciers is indeed unavoidable without a considerable reduction of basal melt, as indicated in our study.

**Code availability**
CISM is an open-source code developed on the Earth System Community Model Portal (EPSCOMB) Git repository available at https://github.com/ESCOMP/CISM. The specific version used to run these experiments is tagged under https://github.com/ESCOMP/CISM/releases/tag/dhdt_version .

**Data availability**
The input dataset, the default initialization and the output of all experiments shown in Table 3 can be found at Zenodo https://zenodo.org/records/13897556 , DOI: 10.5281/zenodo.13897556

**Author contributions**

TvdA designed and executed the main experiments and the sensitivity analysis. WHL and GRL developed CISM and helped configure the model for the experiments. CJB and JAB performed the UFEMISM experiments and processed the data for the sensitivity analysis. RSWvdW and WJvdB provided guidance and feedback. TvdA prepared the manuscript, with contributions from all authors.

TvdA received funding from the NPP programme of the NWO. WHL and GRL were supported by the NSF National Center

for Atmospheric Research, which is a major facility sponsored by the National Science Foundation (NSF) under Cooperative Agreement no. 1852977. Computing and data storage resources for CISM simulations, including the Cheyenne supercomputer (https://doi.org/10.5065/D6RX99HX), were provided by the Computational and Information Systems Laboratory (CISL) at NSF NCAR. GRL received additional support from NSF grant no. 2045075. CJB was supported by PROTECT. This project has received funding from the European Union's Horizon 2020 research and innovation program under grant agreement no.

869304. JB received funding from NWO grant 515.

The authors declare no competing interests.

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
