# Peer review of "Present-day mass loss rates are a precursor for West Antarctic Ice Sheet collapse"

_EGUsphere, 2024_

## Referee Comment (RC2)

Van Akker et al. run two different numerical models to analyze the commitment of the Antarctic ice sheet by initializing with current rates of ice thickness change. Overall, this is a very interesting and worthwhile study. However, I think it requires quite some better framing of the results and more explanation:

1) This method of including observed rates of ice thickness change in the spin-up is not new. Inversion-models such as WAVI and Ua do use observed rates of ice thickness changes (dh/dt) as part of their inversion constraints. The method as presented here is quite ad hoc. It might contain an error – see below. And it is not discussed if the method ever converges, at least the authors do not give any convergence criterion on when they stop their initialization.

2) Claims about existing studies are made, but not backed up by relevant studies. For example, the claim that numerical models struggle to reproduce observed mass change rates accurately has apparently been concluded from the initMIP project (Seroussi et al., 2019). Only, initMIP never assessed this. See more comments in the text. Along these lines, MISI, stability, collapse and similar words are used, but never defined, and in places used incorrectly or misguidingly. Please see detailed comments below.

**Lay Summary:**

Be careful here, you are saying the glaciers will collapse with the timescale of centuries, but you do not mention when that starts. This can be misinterpreted in the press as you stating that "WAIS is collapsing in the next few centuries". Also, your sensitivity tests show a very inconsistent start of this procedure.

**Abstract:**

Line 13: the claim that most models struggle to reproduce observed mass changes is backed up in the introduction using Seroussi et al., 2019. This initMIP paper did however not assess how well models represent current rates of ice loss, neither was this part of the initMIP experiments and aims.

Line 14: I disagree that this initialization method is new.

Line 19-21: I do not think you can conclude this. You are not modelling the physical reason for the current rates of mass loss, but you are forcing them to occur through your initialization method, which is not validated or assessed in any quantifiable way. You are

also not including any feedbacks with climate drivers (e.g., ice-ocean feedbacks) that could enhance or dampen the ice evolution, even for constant climate. You are not considering any effects of variability in climate forcing on the system, and also your GIA initialization is not discussed/validated, so it is unclear how close to reality you are with this.

Moreover, you state "At least a meter in the coming centuries" – what do you mean with coming centuries? Be more specific, it is 2200,2300,2500,... initialization of collapse can be as late as in 2500 years in one of your sensitivity experiments.

**Introduction:**

The introduction needs substantial reworking. There is a misconception of what MISI is, how inversion methods work, and a number of misrepresentations of existing literature and citations of wrong references for statements.

Line 25: This sounds odd as dynamic losses make all of the current mass loss in contrast to surface melting.

Line 26: What are "multiple meters"?

Line 28: You want to credit the original MISI papers here as Pattyn 2018 only summarizes a known concept.

Line 28: What do you mean with "unforced, irreversible retreat"?

Line 29-33: The explanation of MISI is wrong. It is stated that it occurs only for unbuttressed glaciers, and then referred to TWG and PIG as examples of such. But PIG, and to a lesser extent TWG, are not unbuttressed glaciers (e.g., Fürst et al., 2016; Reese et al., 2018). Even more, we would not care about ocean-driven ice shelf melting if they were unbuttressed.

Line 34: Importantly, these papers do not show anything like "already undergoing MISI-like retreat", they only suggest it based on finding continued retreat in their modelling studies (Joughin) or irreversible retreat, but for stronger than current forcing (Favier).

Line 42: How do you define irreversible? How do you define collapse?

Line 42: Importantly, the ABUMIP experiments in Sun et al. 2020 are no projections. They are something like a highly unrealistic worst case, as they remove ice shelves instantaneously and assume that they cannot form again. To use this to conclude that further retreat of PIG and TWG will collapse WAIS over several centuries is wrong. What ABUMIP can maybe provide is a highly unrealistic lower bound on timescales of collapse.

Line 43-45: Importantly, this study (Joughin) does not show anything about irreversible retreat. They discuss retreat. Please make sure to clearly define irreversible retreat, and then read the papers carefully to put that each one in context with your definition of irreversibility.

Line 45: Favier does not show that PIG is currently undergoing irreversible retreat. They show that it would, if forcing is applied. They do not make experiments that analyze the current forcing. What do you mean with "unstable" here – if the retreat could be reversed, is it still unstable?

Line 48: "Several studies suggest that PIG and TWG are unstable under the current climate and could collapse on a timescale of a few centuries." The Joughin study was mentioned before,  Arthern & Williams 2017 make no claims about the glaciers being "unstable current climate" or a collapse within a few centuries. Golledge et al., 2017 does find, coming from the last glacial maximum, an eventual ice loss of much of the marine regions of WAIS under constant current climate. However, they find that after 1500 years constant current climate. Coulon et al., 2023 is an EGU abstract that does not appear to use current climate, but model WAIS under various RCP scenarios.

Line 50: Again, none of these studies supports your claim of "relative stability" (whatever that means). Feldmann & Levermann 2015 do not make statements about stability of the current state of TWG and PIG, rather they show that irreversible retreat, after a perturbation in the ASE, is possible. Arthern & Williams 2017 is the same study you just cited in the sentence before to support the opposite claim, Garbe et al., 2020, again, find hysteresis and that WAIS collapse could occur at around 1 to 2 ∘C of global warming above pre-industrial levels (which is the case at the moment), because they make quasi-equilibrium simulations, they cannot make statements about timescales of collapse. Rosier et al., 2021, does not make any statements about the current state of PIG under current climate conditions, they rather show that 3 tipping points could exist, and Reese et al., does show that WAIS eventually collapses irreversibly under current climate.

Line 54: Garbe does not make a claim about ocean thermal forcing.

Line 57-58: Reese et al., 2023 makes exactly this claim of continued retreat and eventual acceleration due to MISI under present-day climate conditions.

Line 62-63:  The claim that numerical models struggle to reproduce observed mass change rates accurately has apparently been concluded from the initMIP project (Seroussi et al., 2019). Only, initMIP never assessed this.

Line 65 and following. The authors do not seem to have understood how inverse methods work. It is not "data assimilation methods that are used to iterate towards a specific state", and uncertain parameters are not "tuned iteratively". References for the erratic model drift are required. "So far, matching observed mass change is not used as optimization target": actually, the rates of observed rates of ice thickness change have been added also to inversions, for example in WAVI or Ua. Also, the authors miss out on transient inverse calibration (see papers by Dan Goldbergs). They miss out a mix of methods where a similar approach to their own, to modify the surface mass balance during the initialization with a specific target, has been done for example in Hill et al., 2023.

Line 78: Spin-up models do not necessarily aim for their initial state to be close to equilibrium, in particular not when they initialize with a glacial cycle.

Line 83: Naughten 2022 does not analyze trends in historical Amundsen Sea warming in GCMs. Conclusion cannot be drawn. It should be noted that the advantage of using the GCM forcing is that you represent the physical driver of changes, which is something that you do not do.

Line 85: It should be noted that nudging in the spin-up process to obtain observations is not uncommon, this is used in the PSU. Also, the force-to-thickness approach where the surface mass balance is adjusted to obtain present-day geometry is something that has been applied over Greenland in PISM.

**Methods**

Table 1: delta T is supposedly also from Jourdain et al, 2020?

Line 124: When the ice is frozen to the bed, no sliding occurs by definition.

Equation 1.2 This equation is different from the one given in Lipscomb et al. 2021. There, the second term was mentioned to create a dampening, so why do you introduce the last term? I think, actually, the sign of the last term is wrong. If you assume that the first two terms in the brackets are zero, then $C_c > C_r$ would mean that the term becomes positive. Since it's sign is negative, and you multiply with $-C_c$, you end up with $\frac{dC_c}{dt} > 0$, thus nudging your friction parameter further away from the regularization value. Or am I missing something here?

Out of curiosity, why do you have a 2 in the second term? I suppose this is some kind of ad-hoc approach, so why add a 2?

Is there some kind of convergence criterion, or when/how do you decide to stop this nudging process?

Line 135: How do you arrive at these values, why do they make sense?

Equation 1.6: A general comment here is that I think this is not a good idea to tune the $\delta T$ values this way: the way you set them, there is no reason a priori why your values should have any predictive value. In contrast to the sliding parameter, there are some observations and numerical ocean modelling studies available which could be used to inform $\gamma_0$ and $\delta T$ so that you have at least some hope to have the right parameter values. Does the nudging of the sliding parameter not give you enough degrees of freedom to obtain an initial state as you wish?

Additionally, the equation differs from the one in Libscomb et al., 2021. Why did you choose to do this? I think, again, there is a problem with the equation. In the first term, if $H > H_{obs}$ this term is positive, which means that you will decrease $\delta T$ , hence decrease melting, which should thicken the ice further. Similarly, the second term, if $\frac{dH}{dt}$>0, then you will decrease $\delta T$  (we can assume here the case where H close to Hobs), reducing melting and thickening the ice further. Are you sure this is how the equation is intended and implemented?

Is the ocean temperature corrected on a basin-scale, or locally? Make sure to show your final values of $\delta T$ somewhere in the paper.

Line 157: Do you think this choice is justified when comparing with values of $\gamma_0$ from Jourdain et al., 2020?

Line 172: Greene et al., 2022 does not apply a calving law to Antarctic ice shelves, other than removing ice in locations where calving occurred in observations. The Amaral study is for Greenland, you want to have here a reference to an Antarctic study as the calving of ice shelves and tidewater glaciers will be quite different.

Line 182: Refer to the section instead of saying "later".

Line 191: UFEMISM does not use inversion, or? This could be misunderstood, just say "We nudge..." and "UFEMSIM simultaneously nudges ...".

Section 2.2: More detail on the difference between UFEMISM and CISM would be great.

Methods: Please show the nudged fields you use for your simulations somewhere.

Section 2.3: How is the thermodynamics initialized in both models? How is the GIA model initialized if you use one?

Line 204: How do you decide that the method has converged? Is this done for both, CISM and UFEMISM in a similar way?

Line 215: Which dataset do you use for observed mass change rates?

Line 229: In my understanding this is not an inversion.

Line 251: Dow is an EGU abstract. You want to avoid citing non peer-reviewed literature (that is also not accessible) unless really necessary.

Line 259: Swart et al. Is a model intercomparison project description paper, not an original citation for this.

3. Results

Figure 1: The blue boxes on b do now allow to see PIG and TWG grounding lines.

Figure S2: What are the basal melt rates for UFEMSIM?

Line 204: Can you show a map of temperature corrections. What means "on average"? So do you have a local correction in each ice shelf grid cell?

Line 305: I disagree that the velocities are the problem. I think this is simply because you aim to have an equilibrium in a region where in reality ocean temperature drive melting that thins the ice shelf substantially, so you have do dial down the ocean temperatures and reduce melting – in fact, I think this is your motivation to include the observed rates of ice thickness change in the initialization, just looking at it from a different angle?

Importantly, you also never show the fields you generated in your inversions. Also, a figure showing the rates of mass change in present-day compared to observations for both models (currently S4) could be moved to the main text. Your supplementary video would also be useful to have for both models and for longer than 50 years.

**4. Results**

I would not call this "Future states" as you are not making projections.

Line 330: Not sure I understand why you pick this line, and how you say that afterwards "accelerated collapse begins"? In Figure 2c, it looks like there is a similarly high bed peak just upstream of this?

Figure 2c: The x-axis is not "grounding line displacement from t=0", but distance from the grounding line at t=0, or? Otherwise, the red line would be odd.

Line 346: From looking at Fig 2c, it could also be more after 450 to 500 years?

Line 349: Twice the currently modelled value in the same location, or at the grounding line at each point in time? That the ice speeds up upstream when the grounding line retreats would be expected, it would be more interesting to compare velocities at the grounding line during the collapse and before.

Line 254: You only discuss this for TWG, can you also discuss the exact procedure of retreat for PIG?

Line 366: Looking at Figure 8, I do not think you can conclude that the collapse can be halted when switching melting off at all times. Most lines show a slowing ice loss trend (which is not surprising as the forcing is reduced), but the trend seems to still go towards ice loss. I think you would need to model substantially longer time scales, and see that in all cases the ice starts growing again, to make your claim about stopping the collapse.

Line 369: Again, the graph you present is not sufficient to make claims about "stopping the collapse".

Line 371: How do you conclude the MISI resemblance exactly, be a bit more specific. Also, Schoof and Hewitt 2013 is, I think, not an appropriate citation here.

Line 375: This is also shown by for example by Fürst et al., 2016.

Lines 371-379: What do you mean by "pure MISI" - this is only defined for steady states anyways, so would never apply to the real Antarctic ice sheet anyways? As said before, I disagree that your reversal experiments show whether the collapse can be stopped. This whole paragraph needs to be re-considered after extending your simulations, or you can omit it.

Line 382: Holland et al., 2019 does not claim it is natural variability, they find an underlying anthropogenic trend. You state this in the sentence after, so this part seems to contradict itself. Maybe just move the references to the end of the sentence.

Line 387: I find this conclusion a bit odd as basal melting is not directly anthropogenically forced.

Since Figure S8 is discussed quite extensively, I suggest bringing it into the main text.

In this section, a discussion of bedrock uplift would be good, as this is considered one mechanism that could slow down, or halt MISI. How much uplift do you see, how does this compare to estimated rates for WAIS?

**Section 5**

How are these experiments initialized? Do you just change the parameter, or do you run the model again through your initialization procedure with your new set of parameters? How does this affect your experiments?

Line 396: Relative mass loss to what?

Line 401: I disagree, as your melting experiments show that the timescales of retreat depend also on the melt rate applied. They can hence not only be controlled by bed topography or internal system dynamics.

Table 3: Can you add the maximum and average rates of ice loss during "collapse"?

Line 425: Please see introduction about how you can compare these studies. It would be interesting to also compare with Feldmann & Levermann 2015, Reese et al., 2023 mass loss rates here, and analyze differences.

How much does the choice of the dataset of rates of ice thickness change influence your results?

How much would variability in ocean forcing affect the timescales?

A discussion of caveats is missing.

**6. Conclusions**

As mentioned before, your method is not particularly new.

Line 461: Add here your maximum timescale of collapse, and you should mention that this really depends on modelling choices.

Line 464: Except for your lower melt experiments where the phase appears to be longer.

Line 465: What means "irreversible for current climate"? Irreversible is a concept from non-linear dynamics, are you applying this here?

Line 469: I think you want to consider natural variability in ocean conditions and how that influences your argument.

**Data and code availability:**

Consider publishing your scripts and model outputs on a repository such as zenodo.

---

## Author Comment (AC1)

**Reviewer 1**

The manuscript 'Present-day mass loss rates are a precursor for West Antarctic Ice Sheet collapse' by T. van den Akker and co-authors presents a new method to improve the initialization of ice flow models that better captures their current trend and should improve the reliability of projections of future ice sheet evolution. This is a very nice improvement and the method should be applicable to a range of models, therefore having a large impact on this community. The results, based on an ensemble of simulations and the use of two different ice flow models, show that the current mass loss observed in the Amundsen Sea sector is a precursor to large and rapid changes in the future.

I find the idea presented interesting and simple to implement, therefore making it a very nice improvement to other models. However, I find the manuscript confusing at times, some reorganization and changes in the figures/tables would help a lot. I also think the conclusions are not always supported but the results and the manuscript should be clarified to better integrate both models and ensure a direct comparison of the results.

We thank the reviewer for their insightful comments above about the applicability of our method, and the detailed comments below.

Major points:

The current mass loss in West Antarctica is said to be a precursor to large future changes, but there is no direct comparison or quantitative analysis of what would happen otherwise (without the current rate of change) and how it pre-determines the future changes. This should be better justified and quantified, as demonstrated by model results, or this conclusion should be removed from the manuscript.

We thank the reviewer for this comment and understand the confusion about the equilibrium spinup and the spinup using the observed mass change rates, as they are presented in the manuscript. Simulations initialized without the current mass change rates, i.e. the equilibrium spin-up in the manuscript, only develop marginal large mass change rates at all and the WAIS does not collapse. This is because the ice sheet model CISM is designed to behave like that. After an equilibrium spin-up, the model should be stable and in steady state, because we only stop our inversion once our ice thickness does not change anymore, and we therefore cannot get closer to the observed values. So by definition, an equilibrium spin-up results in a modelled ice sheet with dH/dt≈0. Since we employ unforced future simulations to test whether or not the observed mass change rates are sufficient by themselves without climate change to let Thwaites Glacier and Pine Island Glacier collapse, our equilibrium initialization does close to nothing: since dH/dt is close to zero and no forcing has been applied, this is a model drift experiment. A typical model drift simulation with our equilibrium model setup shows a slight thickness increase on the Pine Island glacier, but this model drift is very small compared to the observed and modelled mass change rates (see Fig S6 in the supplementary material).

To make a direct comparison between the equilibrium initialization without dH/dt, and the main experiment with dH/dt, we will add the model drift experiment of the equilibrium initialization (i.e. 1000 years continuation from the equilibrium initialization with fixed inverted fields) as a figure in the supplementary information. Furthermore, we will move the observed and  modelled mass change rates in the first five years of the default experiment to the main text. Additionally, we will show the inverted friction parameter and ocean temperature correction in both initializations in the supplementary material.

Another point is that this new method is used by two different ice flow models, which is great to show its easy implementation and possible impact on a range of models. However, the results from the two models are barely compared and used in a different way. It is therefore difficult to assess if similar results are achieved: for example, showing how spatial mass loss are very different without this method but very similar after including it in both models would show the similarity. In order to quantify the impact of these changes and the improvements made, it would be important to compare the models more directly and systematically, both for the initialization and for the sensitivity experiments.

We thank the reviewer for their positive take on using two ice sheet models. We agree that the results of UFEMISM are not discussed as extensively as we discussed our CISM results. Initially, we developed our method especially for CISM, and when we found the WAIS collapse in all of our forward simulations, we wanted to make sure that this feature was not restricted to CISM before drawing too firm conclusions. For this reason, we added the UFEMISM runs to show that adding dH/dt also for that model shows a collapse. In other words the result is not a CISM artefact. As a secondary goal we hereby show that the method can be easily implemented in another ice sheet model.

We thank the reviewer for the suggestion of including the spatial patterns of mass loss, and we agree that more should be written about the UFEMISM simulations. We will add a figure like Figure 2 in the main text, but with the UFEMISM results, and a section in the results discussing the UFEMISM results in detail.

It is also not clear which experiments have been performed by which model and why such choices were made, and why only the results from CISM are shown most of the time, with barely any result from UFEMISM. This should be clarified to help the readers follow easily the results and conclusions. Similarly, the manuscript is confusing at times and could be made more straightforward (see detailed comments below).

We agree with the reviewer, and as stated above, we will add a section and a figure in the results section showing in detail the results of the main UFEMISM experiment.

Technical comments:

- l.20: it is not clear how the current mass loss is a precursor to future large changes, the mechanism or reason causing this should be mentioned.

We will change this to: 'Our results imply that the thermal forcing present in the present day ocean state, when kept constant, will cause the deglaciation of large parts of the WAIS'

- l.28: MISI is not really uncertain, it is a mechanism reproduced in a number of models and analyzed in depth by many theoretical papers. Its likelihood however is uncertain. This should be rephrased.

We will change this to: 'One such process is the Marine Ice Sheet Instability (MISI; Pattyn et al. (2018)), which could drive the unforced irreversible retreat of marine-terminating glaciers but whose likelihood is uncertain.'

- l.29: 'unbuttressed outlet glaciers' I thought this was applicable to both buttressed and unbuttressed glaciers, even if the rates of fluxes are impacting by the buttressing.

The theoretical concept of MISI, as presented by Schoof 2007 and Schoof 2012, assumes a grounding line without backstress i.e. where there is no buttressing ice shelf. We agree with the reviewer and with Gudmundsson et al 2012 that MISI might occur for buttressed ice shelves as well, and we will remove the discussion on buttressing from this part accordingly.

- it remains unclear whether or not these glaciers are already engaged in an unstable retreat, recent studies suggest it might not have reached such a point yet (Hill, Urruty et al., TC, 2023; Reese, Garbe et al., TC, 2023).

We will add: 'Others suggest that those glaciers have not reached this point yet (Hill et al., 2023; Reese et al., 2023)'

- l.37: higher than what?.

We will change this to 'high'

- l.38: 'a model study' -> 'model studies' (several studies cited).

We will change this to 'model studies'

- l.50-55: another study based on a model ensemble and investigating the marine ice sheet instability mechanism worth discussing here is Robel et al. (2019).

we will add 'Robel et al. (2019) argued that the possibility of MISI amplifies uncertainty in sea level rise projections.'

- l. 60-65: it would be good to also add the uncertainty study by Seroussi et al. (2023).

We will add 'Seroussi et al. (2023) found that the uncertainties in sea level rise projections using an ensemble of ice sheet models increases exponentially with the length of the simulations, with the ice sheet models itself being the main contributor, rather than the forcing.'

- l.67: maybe 'to iterate toward a specific state' -> 'capture conditions at a given time.

We will add 'capture conditions at a given time'

- l.72-73: the problem is rather matching a single time vs a period of change (Goldberg et al., 2015).

We will add 'This can be counteracted locally where (sub)annual observations are available by doing a transient calibration as suggested by Goldberg et al. (2015)'

- l.81: mention that this initialization is done by running very long spin-ups.

We will add 'consists of a long run with'

- l.84: How does that impact the initialization?

It does not impact the initialization directly. What we mean here is that after the initialization, when an ice sheet modeler would want to simulate present-day conditions, they could use the thermal forcing anomaly of an ocean model to simulate the recent warming in the ASE. However, not all ocean models are capable of simulating this warming, so using this output will in those cases lead to underestimating of the melt rates and therefore to an underestimation of the modelled mass loss. We will rewrite this section to clarify these points.

- l.88: Explain the DA biases, which ones and why?

The DA biases targeted here are 1) the pattern mismatch of the thinning rate due to the use of a general cost function and 2) unwanted model drift. 1) is the result of treating the mass change rates error together in the cost function with the velocity error: this will help a modeler to simulate mass change rates overall, but it is virtually impossible to model the observed mass change rates correctly everywhere since there can be a trade off in velocity errors and mass change errors. This error then might become unwanted model drift that might influence the future evolution of the modelled ice sheet. We will rewrite this entire paragraph to make this clear, and to credit the DA method more for applying the mass change rates in another way in their initialization procedure.

- l.90-100: it is not clear when each model is used for the different experiments, when only one model and when both are used? Also why are experiments done with one or both models? I don't know if this is the best place but it should be clarified.

We understand the confusion, and note that the way we framed UFEMISM is flawed. We will make clearer in multiple positions in the manuscript the supportive role of the UFEMISM simulations (described in the major point above).

- l.110: maybe this should go in the UFEMISM section

We agree and will make this change.

- l.133: explain what you mean by `the size of each term'.

We will add 'the relative magnitude of each term compared to the sum'

- l.134: 'linear interpolation'.

We will add this

- l.135: explain why you use these values and how you chose them.

We chose them as they are the default values in PISM. We will add the sentence: 'We chose targets of 0.1 for bedrock below -700 m asl and 0.4 for 700 m asl, with linear interpolation in between, similar to Aschwanden et al. (2013)'

- l.146: why do you use $p = 0.5$? How is that constrained or calibrated?

This is a modelling choice, which is not well constrained by observations. We tested other values between 0 and 1 and found 0.5 to be a reasonable compromise based on inversion freedom (higher p decreases the effectiveness of the friction inversion) and physical intuition (there should be some basal friction decrease for ice resting on a bed far below sea level). We will add: 'A value of p=0.5 is chosen in this study to include some hydrological connection but prevent instabilities during the initialization when using p=1, which was mentioned in the study by Lipscomb et al. (2021) and Leguy et al. (2021).'

- l.155: is this the same \tau or a different o? What is the impact since it is a different physics and orders of magnitude?

The \tau is different. We will distinguish the two variables in the revised text.

- l.169: How is that scaled? Is it done linearly, or bilinearly or something else?

The grounded and floating fractions are determined bilinearly. The method is described by Leguy et al. (2021), which is cited earlier in the paragraph.

- l.171: 'including a few applications' … ' e.g., Yu et al.. (2019)'.

We will change this to 'There are several calving laws in the literature (e.g. (Yu et al., 2019); Wilner et al. (2023); Greene et al. (2022)).'

- l.172: I think it is the same problem for Greenland? Or if not you could explain why it is different between Greenland and Antarctica.

We will change this to: 'However, there is no agreed-upon best approach to Antarctic calving (this holds for the Greenland Ice Sheet as well, see for example Benn et al. (2017)), and most calving laws struggle to reproduce the observed calving front at multiple locations simultaneously (Amaral et al., 2020).'

- l.176: why was 1 m chosen? What would be the impact of choosing a higher threshold?

Conservatively, we did not want to remove too much ice. Since we make conclusions about the collapsing WAIS, we did not want it to be an artifact of a too aggressive calving threshold. A higher limit would decrease buttressing faster and probably increase the rate of collapse. A small threshold ensures that WAIS collapse is caused mainly by sub-shelf melting and not additional calving. We are in the process of developing calving schemes that could be used in a future study to evaluate sensitivity to calving laws.

- l.181: it would be easier if the main experiment had a name or something easy to refer to.

We will refer to it as the'default experiment'. We will change this in the text.

- l.184: the sliding is also different.

Correct, we will note this.

- l.185: 'targeted high resolution': what does that mean?

High resolution only in the places that are dynamically interesting, for example near grounding lines. We will change this to: 'at variable resolution, e.g. simultaneously high resolution at the grounding line and lower resolution in the slow moving interior (Berends et al., 2021).'

- l.197: what else is different between CESM and UFEMISM?

There are several other (small) differences, of which the main difference is the use of triangles instead of rectangles as discretization. Furthermore, UFEMISM uses a different, CFL constraint, time stepping scheme, and a slightly different initialization procedure containing nudging followed by a relaxation time. The latter is absent in CISM.

- l.207: 'with observations like the result of the equilibrium' -> 'with observations, similar to the result of the equilibrium'.

We will change the text as suggested.

- l.210-215: this explication is a bit confusing, since this is a key element of the method, you should add a schematic equation to explain it more clearly.

We will add Eq 1.8 and 1.9, schematic mass balances of an ice sheet featuring a correction term. We will use those schematic equations to, in this paragraph, make more clear what we do to arrive at the observed dH/dt

- l.226: ' would make the ice sheet theoretically more stable': why is it more stable? In your set-up or in the experiments? If you want to show that this method provides more mass loss, what is the best way to demonstrate that? Adding something like that would go a long way.

In the setup without using a dH/dt, we are spinning up to a quasi-steady state in which dH/dt is close to zero. In regions where the observed dH/dt is negative, our inversion will typically compensate by cooling the ocean and/or increasing the basal friction, possibly in ways that are inconsistent with the current ice state. Thus, the spun-up ice sheet is likely to be too stable, and will react differently and possibly slower to forcing. As stated in the reply to the second major point, we think this is a very interesting question and we thank the reviewer for raising it, this will definitely be the scope of future research. Since we focus purely on the effect of the present-day mass change rates in this paper without further forcing. We will rewrite, as stated above as reaction to the first major point, sections 2.2 and 2.3 and clarify the characteristics and difference between the two initializations.

- l.235: is the 'default evolution' similar to a case with constant climate conditions?

Yes. We do not apply forcing, we only let the ice model run with the present-day mass change rates. We do not modify the forcing from the spin-up to the forward run; we let the model run with the same forcing and inversion parameters as obtained during the spin-up, i.e. we run forward from the present-day disequilibrium.

- l.237: in which case is it negligible?

The model drift is negligible, so our forward experiments are not hindered by spurious thinning or thickening. We will change this sentence to: 'Complementary to this default experiment, we verified that the model drift is negligible with forward runs in which the mass balance correction term $CR$ remains included in Eq 1.9 (Fig. S6). In this case, if the spin-up was successful, the modelled ice thickness change rates should be approximately zero'

- l.246: which parameterization is that? Add a reference or equation to explain it.

The PISM parameterization, we will add the same reference (Aschwanden et al 2013) here as well

- Overall I am confused about the forward simulations. The abstract mentioned projections, but I only see the constant climate conditions with and without the mass correction or something like that. I would be important to clarify and describe these experiments with more details.

Thanks for this comment, we noted that our forward simulations are in fact not projections. Since we do not use any forcing, the word projection is confusing, and we thank the reviewer for pointing this out. We will remove the term, and replace it with 'simulations'

- Fig.1: the model used to produce these results should be mentioned at the beginning of the caption. Remind that e and f are observations and which ones were used to calibrate the basal melt for the model.

We will change this accordingly to 'Initialized state of the Antarctic Ice Sheet using CISM'

- l.287: one important method is the difference between the two initialization methods. It should be shown more and not just in the supplementary material. It should also be shown for the two models to compare the impact in both cases and how both methods display similar trends or not with the new initialization.

We will add a figure showing the difference in inverted quantities (Coulomb c and the ocean temperature perturbations) in the equilibrium initialization, to show what the effect of adding the mass change rates is on the inversion. Since the focus of this manuscript is the transient initialization and the continuation run with present day mass change rates, we would like to keep figure S3, the evaluation of the performance of the equilibrium initialization, in the supplementary material. As stated before, there is per definition very little trend in a simulation starting from a successful equilibrium initialization, since we do not change the forcing. The only trend in a simulation starting with the equilibrium initialization is the model drift, and we will add a figure in the supplementary material to show that this is very small, similar to the model drift experiments with the transient initialization.

- l.287: how low is the bias? Add numbers.

We will add this from the figure caption and change this paragraph to: 'The thickness error over the WAIS is low (RMSE = 19.3) , and the GL closely follows observations, with an average 1.5 km difference (calculated as the average distance between the modeled GL position and the closest observed GL position). The modeled GL for PIG is shifted seaward by 5-10 km. The modeled basal melt rates and melt patterns beneath floating ice agree well with the values from Rignot et al. (2013) and Adusumilli et al. (2020), with an integrated melt flux within the range of the two datasets (CISM; integrated flux of 106 Gt/yr, Adusumilli et al. (2020); integrated flux of 94 Gt/yr and Rignot et al. (2013); integrated flux of 149 Gt/yr).'

- l.290: same here, how well does it agree? Add numbers.

See reply to the previous comment

- l.294: are you referring to the models from the ISMIP6 ensemble using data assimilation or spin-up?.

Both, we will add this to the text as 'The root mean square error (RMSE) of the ice surface velocities is comparable to other ISMIP6 models (Seroussi et al., 2020), of which the range is 100 – 400 m/yr, where many are optimized to match observed velocities and use a variety of spin-up and data assimilation methods.'

- l.308: how do these other corrections compare?

When not initializing with dH/dt, our inversion procedure under the ice shelves needs to cool down the ocean more. The assumption of steady state is inconsistent

with the observationally based thermal forcing, which implies high melt rates and thinning. The ocean must cool to reduce the melt rates and achieve a steady-state thickness roughly consistent with observations. In the equilibrium initialization, ocean thermal forcing corrections were largely (on average) negative under Thwaites and Pine Island shelves, they get closer to zero in the transient calibration (in the order of 0.2 – 0.5 K). We will add this to the manuscript.

- Fig.2: why is the grounding line in the Siple Coast and most of Filchner-Ronne not changing at all? How does that compare to previous results?

Because presently there is little to no thinning in these areas, and since we are only applying present-day observed mass change rates, these areas remain stable. We will mention this in the discussion as 'Areas where little to no mass changes are observed in Smith et al. (2020), such as at the Siple Coast and in Victoria Land, remain stable in our simulation'

- l.340: 'sea level change equivalent' . We will add this.

- l.333: what is so special about this ridge? There is another one at about 100 km that is just as high?

In our simulations, this particular ridge is critical for the ice dynamics. Before the glacier reaches this ridge, we see mass loss comparable to the observed rate, but after the glacier ungrounds from this ridge, the mass loss accelerates. The acceleration does not slow when it reaches the ridge at 100 km. This next ridge of 100 km is in this cross section comparable to the first ridge that we identified, but less extended in the direction perpendicular to the cross section: this second ridge is surrounded by deep throughs while the first ridge is a substantial 'bedrock row' stretching almost through the entire Thwaites flow line perpendicular to the flow direction. We will add: 'As soon as the grounding line passes over the line AB in Fig 2A, even the higher ridge at approximately 100 km downstream of the present-day grounding line location cannot stop TG from collapsing. This second ridge is in the cross section of similar height as the ridge AB, but less extended in the cross-flow direction and surrounded by throughs, see Fig S7.'

- l.353: 'a large ice shelf has formed instead': how stable is this ice shelf? How thick/thin is it? I don't remember seeing a mention of what is done for the calving for both models and I am wondering how this result would be impacted by the choice of calving?

Good point. The shelf is about 200 m thick, and the calving front has not moved. A couple of sentences describing the characteristics of this ice shelf will be added: 'A large confined ice shelf has formed instead, with an ice thickness of several hundreds of meters close to the grounding lines, thinning to tens of meters in the direction of the (non-moving) calving front.'

- Fig.3: Again, why just show results from CISM here? If the idea is to look at the similarities between the two models and the possible timeline of collapse, comparing results from the two models would go a long way. It would be

interesting to see if the new initialization method allows to better reconcile results from the two models.

We thank the reviewer for this remark, we will add a more extensive discussion of the results of the UFEMISM runs, also as a reply to earlier comments.

- l.370: are there figures showing these results?

Yes, in the Supplementary material. We will add this figure to the main text, because we refer to it extensively

- l.376: again here, how is that impacting by the choice of calving?

A calving algorithm that moves inward with the thinning ice shelf, with more ice removal, will decrease this buttressing capacity. We will add this explanation to the manuscript.

- l.384: Which former case is discussed here? The previous sentence discussed melting caused in part due to the variability and in part to anthropogenic forcing, so it is a little unclear what former refers to.

The former case refers to the case where natural variability forcing caused the CDW to be present under the shelves. We will reword the sentence.

- l.395: Why is it incorrect? It is not clear if this is just a different evolution and how the "correctness" can be established. What are the conditions or criteria to decide whether or not this is correct.

We agree that the word 'correct' is not suitable in this context. We will remove the word 'correct' .

- l.396: "set of model choices": which choices are being considered? There should be a clear list of choices and parameters.

The model choices are shown in Table 1, we will add a reference to that table here.

- l.410: "weaker" and "stonger" should be discussed and quantified. What parameters are changed?

This is gamma_0 in Eq 1.5, we will add the reference here.

- l.414: by how much is it delaying the collapse?

Several centuries, we will add the exact timing.

- l.415: where are these results shown?

We will show these in the new UFEMISM paragraph and figure.

- l.421: in what sense is "linear" used here? Is it the same rate of retreat? Or evolution of grounded/floating areas? And which parameters impact it?

We agree that 'linear' is not the right word in this context. We mean that the retreat rate is similar to what is observed today. We will clarify this point.

- Fig.4: why are only 2 results from UFEMISM shown on this figure? Also, would there be a way to organize the runs by colors and symbols according to parameters changed or something more intuitive? It is a bit difficult to read in this format. The legend mentions some broad categories but is not very clear (e.g. what does forcing refer to since there is also a basal melt category, which is a big part of the forcing).

UFEMISM will have its own paragraph where we also make clear that CISM is the main model used for this analysis and to develop this method, and UFEMISM is the model to show that the method works in another model as well, and that the WAIS collapse is not a CISM-only feature. We thank the reviewer for the suggestion on the formatting, we tried several ways of showing these results and judged this way to be the most visible one. We will add a description of which runs falls in which category.

- l.486: what about the availability of the code to reproduce the results presented and reproduce the figures presented in this manuscript?

This will be uploaded.

- Supplement movie: Clarify time in supplement movie (e.g., 10001.0, not clear what this time corresponds to).

We will change this movie and reassess its added value to the manuscript.

---

## Author Comment (AC2)

**Reviewer 2**

Van Akker et al. run two different numerical models to analyze the commitment of the Antarctic ice sheet by initializing with current rates of ice thickness change. Overall, this is a very interesting and worthwhile study. However, I think it requires quite some better framing of the results and more explanation:

1) This method of including observed rates of ice thickness change in the spin-up is not new. Inversion-models such as WAVI and Ua do use observed rates of ice thickness changes (dh/dt) as part of their inversion constraints. The method as presented here is quite ad hoc. It might contain an error – see below. And it is not discussed if the method ever converges, at least the authors do not give any convergence criterion on when they stop their initialization.

We thank the reviewer for their insightful comments about the novelty of this method. We agree that our proposed method is not completely new, and that especially in the Data Assimilation models dH/dt has longer been used in the cost functions of those models. We thank the reviewer for the suggestions regarding UA and WAVI, and we will reframe our method from being new to being an addition to models using the spinup initializiation, in order for those models to be able to represent the observed mass change rates in their initialization method. We will furthermore give credit to the DA models that do use de mass change rates in their initialization. We will rewrite the corresponding parts in the introduction, specifically lines 65 – 85 thoroughly.

We can assure the reviewer that our method converges, and this is indeed implied, but never explicitly mentioned in the main text. We thank the reviewer for pointing this out. Our model drift experiments act as a convergence criterion: as soon as we stop our inversion and keep the inverted fields constant, our model shouldn't drift (or at least, very little). We demonstrate that there is little drift, especially compared to the modelled mass change rates, in the supplementary information (Figures S5 and S6). To us, this is our convergence check, but we will add this to the main text as well, with an explicit discussion on our model convergence and what the model drift means for our future simulations.

2) Claims about existing studies are made, but not backed up by relevant studies. For example, the claim that numerical models struggle to reproduce observed mass change rates accurately has apparently been concluded from the initMIP project (Seroussi et al., 2019). Only, initMIP never assessed this. See more comments in the text. Along these lines, MISI, stability, collapse and similar words are used, but never defined, and in places used incorrectly or misguidingly. Please see detailed comments below.

We thank the reviewer for their sharpness and we will adjust the references where needed, following the comments below. We apologize for any misguiding and recognize the sloppiness in our references. We will adjust the manuscript according to the detailed remarks.

**Lay Summary:**

Be careful here, you are saying the glaciers will collapse with the timescale of centuries, but you do not mention when that starts. This can be misinterpreted in the press as you stating that "WAIS is collapsing in the next few centuries". Also, your sensitivity tests show a very inconsistent start of this procedure.

We cannot access the lay summary anymore, we will fix this with the editor

**Abstract:**

Line 13: the claim that most models struggle to reproduce observed mass changes is backed up in the introduction using Seroussi et al., 2019. This initMIP paper did however not assess how well models represent current rates of ice loss, neither was this part of the initMIP experiments and aims.

We will rewrite the framing of the initialization methods in the introduction entirely following major point 1. We will adjust the abstract accordingly, and add more proper references in the introduction (like Aschwanden et al 2019)

Line 14: I disagree that this initialization method is new.

We agree, and we will change the framing of our method throughout the manuscript

Line 19-21: I do not think you can conclude this. You are not modelling the physical reason for the current rates of mass loss, but you are forcing them to occur through your initialization method, which is not validated or assessed in any quantifiable way. You are

also not including any feedbacks with climate drivers (e.g., ice-ocean feedbacks) that could enhance or dampen the ice evolution, even for constant climate. You are not considering any effects of variability in climate forcing on the system, and also your GIA initialization is not discussed/validated, so it is unclear how close to reality you are with this.

We agree with the reviewer on this point, we need to be very clear about the limitations of our setup and what that implies for the representation of a plausible future evolution. First, since we are not applying any forcing, we are not doing (realistic) projections, but rather realizations or future simulations. We will change this throughout the manuscript (i.e. 'projections' to 'simulations'). Second, we validate our initialization method against observed ice thickness, ice surface velocities and basal melt rates, but as stated in the major points, we do not discuss model convergence (even though our model converges). We will clarify this in the manuscript, in paragraph 2.3 about our initialization method. We will also include figures on the model drift in the equilibrium initialization as well. Thirdly, we are indeed obtaining possibly unphysically mass change rates through our initialization and we will mention this in the main text. Fourth, to a first order there are ice-ocean feedbacks in our simulations (melt rate – depth feedback: lower ice draft will be in deeper and warmer waters with higher basal melt rates, which will increase the ice draft) and we tested what would happen if the mass change rates would be half of todays rates (which is an estimate of what natural variability could cause the mass change rates to be by chance during the observations), and we added GIA in some of our sensitivity tests (without initializing, i.e. assuming wrongly that the earth is in equilibrium with the present day ice sheet) and our mass change rates are GIA corrected. All these points mentioned will be added to the manuscript.

Lastly, we will frame our results from a general point of view 'WAIS will likely collapse with present day mass change rates' to 'Our results imply that the thermal forcing present in the present-day ocean, when kept constant, will cause the deglaciation of large parts of the WAIS'.

Moreover, you state "At least a meter in the coming centuries" – what do you mean with coming centuries? Be more specific, it is 2200,2300,2500,… initialization of collapse can be as late as in 2500 years in one of your sensitivity experiments.

We will change this sentence to be less bold. As the previous remark and reply highlighted: we should be clear on the fact that our setup is limited to the ice dynamical response to current mass change rates, and that this is no realistic projection. We will change this to: 'After 2100, however, dynamical processes are highly uncertain, possibly accelerating GMSL rise significantly, which is in line with (Fox-Kemper et al., 2021; Payne et al., 2021).'

**Introduction:**

The introduction needs substantial reworking. There is a misconception of what MISI is, how inversion methods work, and a number of misrepresentations of existing literature and citations of wrong references for statements.

We thank the reviewer, and we will rewrite the introduction according to the major points, and the detailed comments below.

Line 25: This sounds odd as dynamic losses make all of the current mass loss in contrast to surface melting.

We will changed this to 'Dynamical processes are not expected to accelerate or decelerate significantly before 2100.'

Line 26: What are "multiple meters"?

We cannot state a single clear number here so we will rephrase

Line 28: You want to credit the original MISI papers here as Pattyn 2018 only summarizes a known concept.

Good point, we will add the references of Schoof 2007 and Schoof 2012 here.

Line 28: What do you mean with "unforced, irreversible retreat"?

We mean self-sustaining, that the glacier, given a certain climate and an initial perturbation, will move towards a new, much smaller initial state, to which it will equilibrate. The initial (small) negative of the perturbation will not be enough to move the glacier back to its old position. We will rephrase these two sentences to include this

We will rewrite these two sentences (also as a reply to the previous comment), to: 'One of such processes is the Marine Ice Sheet Instability (MISI; see Schoof (2012, 2007) and Durand et al. (2009)), which could drive the unforced irreversible retreat of marine-terminating glaciers but whose likelihood is uncertain. This process could drive the irreversible (i.e. the glacier will not return to its present grounding line position when the forcing is removed) and self-sustaining retreat of marine-terminating glaciers'

Line 29-33: The explanation of MISI is wrong. It is stated that it occurs only for unbuttressed glaciers, and then referred to TWG and PIG as examples of such. But PIG, and to a lesser extent TWG, are not unbuttressed glaciers (e.g., Fürst et al., 2016; Reese et al., 2018). Even more, we would not care about ocean-driven ice shelf melting if they were unbuttressed.

We note that PIG is indeed (heavily) buttressed, and removed it as example here. TWG is believed to be significantly less buttressed (or not at all). We will add 'Thwaites Glacier (TG) and Pine Island Glacier (PIG), although buttressed'. We will add that MISI is also relevant for buttressed glaciers, and add the reference of Gudmundsson et al 2012 to this part.

Line 34: Importantly, these papers do not show anything like "already undergoing MISI-like retreat", they only suggest it based on finding continued retreat in their modelling studies (Joughin) or irreversible retreat, but for stronger than current forcing (Favier).

We will change this to: 'It is suggested that those glaciers are experiencing continued, accelerated retreat in future simulations and might unground completely in the next centuries (Joughin et al., 2014; Favier et al., 2014; Seroussi et al., 2017). Others suggest that those glaciers have not reached this point yet (Hill et al., 2023; Reese et al., 2023)'

Line 42: How do you define irreversible? How do you define collapse?

We will adopt the definition of irreversible from the IPCC glossary: 'A perturbed state of a

dynamical system is defined as irreversible on a given timescale, if the recovery time scale from this state due to natural processes is substantially longer than the time it takes for the system to reach this perturbed state.' Collapse: fast deglaciation (not quantified) leading to an almost complete loss of grounded ice in a basin. We will clarify this

Line 42: Importantly, the ABUMIP experiments in Sun et al. 2020 are no projections. They are something like a highly unrealistic worst case, as they remove ice shelves instantaneously and assume that they cannot form again. To use this to conclude that further retreat of PIG and TWG will collapse WAIS over several centuries is wrong. What ABUMIP can maybe provide is a highly unrealistic lower bound on timescales of collapse. We agree that ABUMIP is not a suitable reference is this context and we will remove this from the manuscript

Line 43-45: Importantly, this study (Joughin) does not show anything about irreversible retreat. They discuss retreat. Please make sure to clearly define irreversible retreat, and then read the papers carefully to put that each one in context with your definition of irreversibility.

We will remove the word 'irreversible', and explain what we mean by it as a reaction to an earlier comment

Line 45: Favier does not show that PIG is currently undergoing irreversible retreat. They show that it would, if forcing is applied. They do not make experiments that analyze the current forcing. What do you mean with "unstable" here – if the retreat could be reversed, is it still unstable?

To avoid confusion, we will remove 'unstable' and 'irreversible', we reread the Favier paper and reformulated. We will reformulate this sentence to the sentence placed four comments before.

Line 48: "Several studies suggest that PIG and TWG are unstable under the current climate and could collapse on a timescale of a few centuries." The Joughin study was mentioned before, Arthern C Williams 2017 make no claims about the glaciers being "unstable current climate" or a collapse within a few centuries. Golledge et al., 2017 does find, coming from the last glacial maximum, an eventual ice loss of much of the marine regions of WAIS under constant current climate. However, they find that after 1500 years constant current climate. Coulon et al., 2023 is an EGU abstract that does not appear to use current climate, but model WAIS under various RCP scenarios.

We apologize for this sloppy part regarding our references, and we would like to thank the reviewer for pointing this out and their suggestions. We will thoroughly reread the references used here and rephrase and reorder this paragraph accordingly (and change the EGU abstract to the published paper) and reread Arthen C Williams 2017. We will add the timescale to Golledge et al 2017, We will change Coulon from the EGU abstract to the actual paper.

Line 50: Again, none of these studies supports your claim of "relative stability" (whatever that means). Feldmann C Levermann 2015 do not make statements about stability of the current state of TWG and PIG, rather they show that irreversible retreat, after a perturbation in the ASE, is possible. Arthern C Williams 2017 is the same study you just cited in the sentence before to support the opposite claim, Garbe et al., 2020, again, find hysteresis and that WAIS collapse could occur at around 1 to 2 °C of global warming above pre-industrial levels (which is the case at the moment), because they make quasi-equilibrium simulations, they cannot make statements about timescales of collapse. Rosier et al., 2021, does not make any statements about the current state of PIG under current climate conditions, they rather show that 3 tipping points could exist, and Reese et al., does show that WAIS eventually collapses irreversibly under current climate.

We understand the confusion of the reviewer, apologize again for our sloppiness and thank the reviewer again for their sharp eye and we will rewrite this paragraph accordingly. We will change this paragraph to:

'Several modelling studies have assessed the potential for ASE collapse. One study (Joughin et al., 2014) argued that under present-day melt rates, TG might already be on a trajectory toward accelerated retreat; moderate retreat in this century will likely be

followed by a phase of rapid collapse beginning in the next 200 to 900 years. Another study (Favier et al., 2014) used three ice sheet models to show that PIG is now undergoing a forced 40-km retreat (but makes no projections after this). The retreat could be reversed by sufficient ocean cooling (Favier et al., 2014). Both studies mention MISI as the main driver of retreat. Subsequently, studies suggest that TG and PIG are unstable under the current climate and could collapse on a timescale up to 2000 years (Golledge et al., 2021; Coulon et al., 2024) and others suggest that the two glaciers will collapse with additional forcing (Feldmann and Levermann, 2015; Arthern and Williams, 2017; Reese et al., 2023; Garbe et al., 2020). '

Line 54: Garbe does not make a claim about ocean thermal forcing.

We agree, and we will remove the reference here.

Line 57-58: Reese et al., 2023 makes exactly this claim of continued retreat and eventual acceleration due to MISI under present-day climate conditions.

We will add Reese et al, 2023

Line 62-63:  The claim that numerical models struggle to reproduce observed mass change rates accurately has apparently been concluded from the initMIP project (Seroussi et al., 2019). Only, initMIP never assessed this.

We agree and will add Aschwanden et al 2021 here and rephrase to :

'Seroussi et al. (2023) found that the uncertainties in sea level rise projections using ice sheet model increases exponentially with the length of the simulations, with the ice sheet models itself being the main contributor. Uncertainties in ice sheet modelling arise from four main sources according to Aschwanden et al. (2021): suboptimal ice sheet model initialization, incomplete physical understanding of important processes, numerical model uncertainty, and uncertainty in the climate forcing. With respect to initialization, some ice sheet models employing the so-called spin up initialization method, struggle to represent observed present-day mass change rates, because the inherent result of a successful initialization is a stable ice sheet without model drift. Representing these decadal-long present-day mass change rates right is essential for reliable projections, as these changes are the primary observable of the dynamic state of the ice sheet.'

Line 65 and following. The authors do not seem to have understood how inverse methods work. It is not "data assimilation methods that are used to iterate towards a specific state", and uncertain parameters are not "tuned iteratively". References for the erratic model drift are required. "So far, matching observed mass change is not used as optimization target": actually, the rates of observed rates of ice thickness change have been added also to inversions, for example in WAVI or Ua. Also, the authors miss out on transient inverse calibration (see papers by Dan Goldbergs). They miss out a mix of methods where a similar approach to their own, to modify the surface mass balance during the initialization with a specific target, has been done for example in Hill et al., 2023.

We will change 'iterate towards a specific state' to 'capture conditions at a certain time'. We will change 'tuned iteratively' to 'minimize a cost function between observed and modelled quantities'. We will remove the sentence about model drift if we cannot find proper references. We will remove the sentence about optimization targets and will add a discussion on the methods used in UA, WAVI, and by Dan Goldberg and Emily Hill. We will add the following paragraph:

'The dynamic state does not necessarily resemble the observed mass change rates, even when using a cost function to iterate towards these rates. This can be counteracted locally where (sub)annual observations are available by doing a transient calibration as suggested by Goldberg et al. (2015). Also, the modelled mass change rates can be added to the cost function as was done for example by Rosier et al. (2021) in the ice sheet model Úa (see e.g. Gudmundsson et al. (2012)) to minimize model drift, by penalizing non-zero ice thickness change rates in the same way as was done for the ice surface velocity mismatch. Bett et al. (2023) used the ice sheet model WAVI (Arthern et al., 2015) and added the mass change rates from Smith et al. (2020) into the cost function but now penalizing model drift difference with the observed mass change rates. The WAVI inversion method described in Arthern et al. (2015) results in a grid based value for two free parameters relating to basal friction and the ice viscosity. Since both ice velocities and mass change rates are targeted simultaneously through the cost function, it is likely that there is a trade-off between errors in the two target quantities and that the modelled mass change rates do not necessarily agree with observations in all locations. A quantitative comparison between modelled and observed mass change rates is missing in Arthern et al. (2015) and Bett et al. (2023).'

Line 78: Spin-up models do not necessarily aim for their initial state to be close to equilibrium, in particular not when they initialize with a glacial cycle.

We will adjust the text accordingly and clarify that we mean the present-day state of the ice sheet, and credit spin-up models

Line 83: Naughten 2022 does not analyze trends in historical Amundsen Sea warming in GCMs. Conclusion cannot be drawn. It should be noted that the advantage of using the GCM forcing is that you represent the physical driver of changes, which is something that you do not do.

We will remove the reference and agree with the reviewer that the physical driver of change when using GCM forcing is valuable. Our method is a trade-off between modelling the physical driver of change and having full control on where the numerical same mass change rates as observed will be applied. Although we tune ocean temperatures to fit with mass change rates, this is completely different from using full GCM output.

Line 85: It should be noted that nudging in the spin-up process to obtain observations is not uncommon, this is used in the PSU. Also, the force-to-thickness approach where the surface mass balance is adjusted to obtain present-day geometry is something that has been applied over Greenland in PISM.

We do not claim that the nudging method is uncommon. We are not familiar with the 'force-to-thickness approach', and not with 'PSU'.

**Methods**

Table 1: delta T is supposedly also from Jourdain et al, 2020?

No, in our study, this is an inversion targeted variable. We will make this clearer in the table.

Line 124: When the ice is frozen to the bed, no sliding occurs by definition.

Correct, thanks for the remark. We will change this to 'When the ice slides over rough and non-deformable hard beds'.

Equation 1.2 This equation is different from the one given in Lipscomb et al. 2021. There, the second term was mentioned to create a dampening, so why do you introduce the last term? I think, actually, the sign of the last term is wrong. If you assume that the first two terms in the brackets are zero, then $Cc > Cr$ would mean that the term becomes positive. Since it's sign is negative, and you multiply with $-Cc$, you end up with $dCc > 0$, thus nudging your friction parameter further away from the regularization value. Or am I missing something here?

That is correct! Thanks, we changed the minus to a plus. This was wrongly copied over from the code. This equation is different compared to the earlier CISM paper because there have been updates in the code. The second term is still a dampening term that, during runtime, will decrease the change in the friction parameter if the ice thickness is already moving into the right direction. The third term is a relaxation term that 'pulls' the tunable parameter away from being over tuned to a modelers chosen target.

Out of curiosity, why do you have a 2 in the second term? I suppose this is some kind of ad- hoc approach, so why add a 2?

No particular reason, this evolved from an earlier version of the CISM code. Since there is a parameter H_0 in this term, it does not have added value. Removing it and doubling H_0 would have the same effect.

Is there some kind of convergence criterion, or when/how do you decide to stop this nudging process?

When the ice sheets exhibit very little to no model drift when the inversion is stopped and the model is allowed to run forward with the inverted fields. This is achieved, and we will clarify better in the text that we tested this. We will add this to section 2.2, and also quantify this for the equilibrium initialization in the supplementary material.

Line 135: How do you arrive at these values, why do they make sense?

Generally, we follow the logic that deeper beds should have weaker till. There might have been changes historically between ice covered and ocean covered, creating a weaker till layer. We took the exact values from PISM. A reference will be added, Aschwanden et al (2013).

Equation 1.6: A general comment here is that I think this is not a good idea to tune the $\delta T$ values this way: the way you set them, there is no reason a priori why your values should have any predictive value. In contrast to the sliding parameter, there are some observations and numerical ocean modelling studies available which could be used to inform $\gamma_0$ and $\delta T$ so that you have at least some hope to have the right parameter values. Does the nudging of the sliding parameter not give you enough degrees of freedom to obtain an initial state as you wish?

It does not, if we want the grounding line to be at the correct position (which we definitely want when doing stability analysis) we need a free parameter related to basal melting as well. We agree to the weakness the reviewer points out regarding the predictive value of delta T, and we note that the corrections needed are often small and average out over an ice shelf to something close to zero. However, since our basal melt scheme is based on a single parameterization (rather than a sub model, or a coupled ice-ocean model) we think it is justified to allow some tuning here, since the parameterization itself in combination with the dataset from Jourdain et al (2020) will not provide us with the right basal melt rates. Typically the tuning is within the accuracy of the dataset itself. We agree with the reviewer that the deltaT tuning is somewhat arbitrary and will make this more clear in the paper. We will add to paragraph 2.1.2:
'In a forward run, we keep $\delta T$ constant. When the grounding line retreats, we interpolate basin-average (basins as identified by Zwally et al. (2015)) values of $\delta T$ newly floating cells. Full interactive Ice-ocean coupling using CISM combined with an ocean model is under development. '

Additionally, the equation differs from the one in Libscomb et al., 2021. Why did you choose to do this? I think, again, there is a problem with the equation. In the first term, if $H > H_{obs}$ this term is positive, which means that you will decrease $\delta T$, hence decrease melting, which should thicken the ice further. Similarly, the second term, if $\frac{dH}{dt} > 0$, then you will decrease $\delta T$ (we can assume here the case where H close to Hobs), reducing melting and thickening the ice further. Are you sure this is how the equation is intended and implemented?

Thanks, we removed the minus in front of the deltaT, which is erroneous. In addition, this equation differs for the same reason why we changed our friction inversion as it proved to make the convergence during the initialization faster. We will add a sentence to both equations stating this. We will add:

'This equation is slightly different compared to Lipscomb et al. (2021), as we found that this equation yields more accurate results in terms of ice thickness, and converges faster.'

Is the ocean temperature corrected on a basin-scale, or locally? Make sure to show your

final values of $\delta T$ somewhere in the paper.

We agree to show the delta T values, as well as our inverted coulomb c, they will be added to the supplementary online material in a four panel figure: coulomb c and delta T for the whole AIS and for the ASE region, for both the transient and the equilibrium initialization.

Line 157: Do you think this choice is justified when comparing with values of $\gamma_0$ from Jourdain et al., 2020?

Yes, this value for the non-local slope parameterization is in the same order of magnitude as the values presented there, and what is more, this has been tested specifically for CISM in the Lipscomb et al 2021 paper. We will add this reasoning to the text.

Line 172: Greene et al., 2022 does not apply a calving law to Antarctic ice shelves, other than removing ice in locations where calving occurred in observations. The Amaral study is for Greenland, you want to have here a reference to an Antarctic study as the calving of ice shelves and tidewater glaciers will be quite different.

We will add a discussion on Antarctic calving in the text as well. We will add: 'There are several calving laws in the literature (e.g. (Yu et al., 2019); Wilner et al. (2023); Greene et al. (2022)). However, there is no agreed-upon best approach to Antarctic calving (this holds for the Greenland Ice Sheet as well, see for example Benn et al. (2017)), and most calving laws struggle to reproduce the observed calving front at multiple locations simultaneously (Amaral et al., 2020).'

Line 182: Refer to the section instead of saying "later".

We will adapt this to: 'We use the same transient initialization procedure including the mass change rates from Smith et al. (2020) in the mass balance equation like in Eq 1.9'

Line 191: UFEMISM does not use inversion, or? This could be misunderstood, just say "We nudge…" and "UFEMSIM simultaneously nudges …".

UFEMISM does use inversion, we thank the reviewer for this suggestion, and we will use this in the main text

Section 2.2: More detail on the difference between UFEMISM and CISM would be great.

We will add a separate UFEMISM section in the manuscript and expand this part of the methods to include a better UFEMISM description here.

Methods: Please show the nudged fields you use for your simulations somewhere.
We will add those to the supplementary materials, and discuss the difference in inverted fields between the mass change rates run and the former initialization in the main text

Section 2.3: How is the thermodynamics initialized in both models? How is the GIA model initialized if you use one?

Thermodynamics: with the Robin solution. GIA: assumes that the bed is in steady state at t=0. This is not necessarily realistic but the best we can do with CISM right now (and with, many other models). Both statements will be added to the text like: 'The thermodynamic profile is initialized with the Robin solution, and no GIA is used in our default experiments i.e. we assume a static bedrock.'

Line 204: How do you decide that the method has converged? Is this done for both, CISM and UFEMISM in a similar way?

By assessing the modeldrift when the inversion is stopped. This is done in a similar way in both models. This explanation will be added to the text. We will add: 'This typically takes 10^4 model years for the model to converge, which we assess by calculating the model drift left in the modelled ice sheet when stopping the inversion (i.e. by continuing the simulation with the inverted fields kept constant). If the model drift is about two orders of magnitude smaller than the observed and modelled mass change rates, and there is no spurious grounding line movement in key regions (like PIG and TG), we accept the initialization'

Line 215: Which dataset do you use for observed mass change rates?
Smith et al 2020, reference will be added

Line 229: In my understanding this is not an inversion.

In our understanding, our nudging procedure is an inversion. We will rephrase it to 'Stopping the nudging procedure and keeping the nudged fields constant.

Line 251: Dow is an EGU abstract. You want to avoid citing non peer-reviewed literature (that is also not accessible) unless really necessary.
We will remove this citation .

Line 259: Swart et al. Is a model intercomparison project description paper, not an original citation for this.

We will change this sentence to: 'For example, cooling of the Southern Ocean sea surface (Bintanja et al., 2015) and reduced Antarctic Bottom Water formation (Williams et al., 2016).'

3. Results

Figure 1: The blue boxes on b do now allow to see PIG and TWG grounding lines.

We will make them smaller.

Figure S2: What are the basal melt rates for UFEMSIM?

We will add a UFEMISM discussion like the discussion of the CISM initialization

Line 204: Can you show a map of temperature corrections. What means "on average"? So do you have a local correction in each ice shelf grid cell?

Yes, we will do this in the supplementary material for both the equilibrium and transient initialization. On average means on average per shelf, so we calculated both for Thwaites and Pine Island separately what the average ocean correction is, since the inversion is grid cell based. We will clarify this

Line 305: I disagree that the velocities are the problem. I think this is simply because you aim to have an equilibrium in a region where in reality ocean temperature drive melting that thins the ice shelf substantially, so you have do dial down the ocean temperatures and reduce melting – in fact, I think this is your motivation to include the observed rates of ice thickness change in the initialization, just looking at it from a different angle?

Correct, our wording in this paragraph is fuzzy on that point, but this is indeed the point we tried to make. We thank the reviewer for this remark, and we will rephrase this. We will add: 'In the equilibrium initialization, ocean temperatures in the Amundsen Basin on average must be reduced by ~0.5 K compared to the thermal forcing dataset from Jourdain et al. (2020)  to reproduce the observed ice shelf geometry. In the observations of Smith et al. (2020), the shelves are thinning rapidly. Simulating large negative mass change rates here requires higher ocean temperatures.'

Importantly, you also never show the fields you generated in your inversions. Also, a figure showing the rates of mass change in present-day compared to observations for both models (currently S4) could be moved to the main text. Your supplementary video would also be useful to have for both models and for longer than 50 years.

We will add figure S4 to the main text. We will add fields of inverted coulomb c and delta T of both initializations to the supplementary material. We will rethink the need for a supplementary video and if we will keep it, we will extend the length.

**4. Results**

I would not call this "Future states" as you are not making projections.

We will change it to 'realizations'

Line 330: Not sure I understand why you pick this line, and how you say that afterwards "accelerated collapse begins"? In Figure 2c, it looks like there is a similarly high bed peak just upstream of this?
Before the glacier reaches this ridge, we see linear mass loss comparable to the observed rate and accelerated mass loss when the glacier ungrounds here, despite the ridge of 100 km. This identified ridge is the 'last resort' for Thwaites glacier: as soon as the grounding line passes this one, the accelerated collapse happens. We will add this explicitly to the manuscript as: As soon as the grounding line passes over the line AB in Fig 2A, even the higher ridge at approximately 100 km downstream of the present day grounding line location cannot stop TG from collapsing. This second ridge is in the cross section of similar height as the ridge AB, but less extended in the cross-flow direction and surrounded by throughs, see Fig S7.

Figure 2c: The x-axis is not "grounding line displacement from t=0", but distance from the grounding line at t=0, or? Otherwise, the red line would be odd.

We will change this accordingly.

Line 346: From looking at Fig 2c, it could also be more after 450 to 500 years?

We will change this accordingly as well

Line 349: Twice the currently modelled value in the same location, or at the grounding line at each point in time? That the ice speeds up upstream when the grounding line retreats would be expected, it would be more interesting to compare velocities at the grounding line during the collapse and before.

The grounding line constantly moves so this would make the comparison difficult. But we will assess the velocities along some flow lines following the grounding line to see if we can find doubling there as well. We will change the text accordingly to: 'Ice velocities in the main channel of TG exceed 4000 m/yr, about twice the current modeled values at that same location (Fig. 3e).'

Line 254: You only discuss this for TWG, can you also discuss the exact procedure of retreat for PIG?
In our simulations, PIG is 'attacked from the side by TG'. So TG collapses and takes PIG with it. We therefore argue that it is more interesting to look at the evolution of TG in more detail, since a collapsing TG takes PIG with it. We will clarify this in the text as: 'PIG is 'dragged along' by a collapsing TG. Both glaciers also collapse when the observed mass change rates are only applied to their single basins (not shown). '

Line 366: Looking at Figure 8, I do not think you can conclude that the collapse can be halted when switching melting off at all times. Most lines show a slowing ice loss trend (which is not surprising as the forcing is reduced), but the trend seems to still go towards ice loss. I think you would need to model substantially longer time scales, and see that in all cases the ice starts growing again, to make your claim about stopping the collapse.

In all experiments where the basal melt is switched off (yellow lines in Figure S8a, the ice sheet starts to regrow. We will add: 'We observe two features in these 'Zero basal melting' experiments: the ice sheet never grows back with the same rate as it collapsed, and the grow back rate is lower when the basal melt rates are switched off later during the simulation.'

Line 369: Again, the graph you present is not sufficient to make claims about "stopping the collapse".
We will change this to: ' We furthermore tested the effect of an instant cool-down at 250 years (brown lines) and 500 years (red lines) during the simulation. None of the cool-down experiments where enough to regrow the ice sheet. We furthermore tested a percentual decrease in basal melt rates at certain timesteps. Both 25% and 50% decrease (yellow and brown lines) in basal melt rates are not enough to stop the collapse, only to slow it down. A 75% (red lines) decrease in basal melt rates turns the mass change rates from negative to eventually positive, but only when applied before 200 years in the simulation.'

Line 371: How do you conclude the MISI resemblance exactly, be a bit more specific. Also, Schoof and Hewitt 2013 is, I think, not an appropriate citation here.

Since there is accelerated collapse on a retrograde bedslope in all our simulations, we assess the collapse to be at least MISI – like. We cannot show, like in Schoof et al 2007 (which is the reference we will use here), mathematically that this is the same instability as in the paper. We will clarify this in this paragraph.

Line 375: This is also shown by for example by Fürst et al., 2016. We will read this reference and add it to the text.

Lines 371-379: What do you mean by "pure MISI" - this is only defined for steady states anyways, so would never apply to the real Antarctic ice sheet anyways? As said before, I disagree that your reversal experiments show whether the collapse can be stopped. This whole paragraph needs to be re-considered after extending your simulations, or you can omit it.

We will remove the word 'pure', because we agree with the reviewer that the theoretical MISI formulation of Schoof 2007 is not applicable to most real life cases, since there are multiple differences with the theory presented by Schoof (to start, this is a 3D case, and the grounding line is not necessarily (back)stress free. We tried to show that, when the forcing is completely removed (in this case the ocean stops melting the ice shelves) the collapse does not continue. In the theoretical case of Schoof, MISI is not influenced at all by (ocean) melt and is self-sustained on retrograde beds since the GL flux will always increase. We will rewrite this paragraph to clarify.

Line 382: Holland et al., 2019 does not claim it is natural variability, they find an underlying anthropogenic trend. You state this in the sentence after, so this part seems to contradict itself. Maybe just move the references to the end of the sentence.

We will do this.

Line 387: I find this conclusion a bit odd as basal melting is not directly anthropogenically forced.

We will add 'via the ocean temperatures'

Since Figure S8 is discussed quite extensively, I suggest bringing it into the main text.
We will do this.

In this section, a discussion of bedrock uplift would be good, as this is considered one mechanism that could slow down, or halt MISI. How much uplift do you see, how does this compare to estimated rates for WAIS?

we will add a paragraph on GIA, the rates in CISM, its effect on MISI and some observations in our missing processes paragraph, and at the beginning of section 5.

**Section 5**

How are these experiments initialized? Do you just change the parameter, or do you run the model again through your initialization procedure with your new set of parameters? How does this affect your experiments?

Depending on the parameter we either rewrite the necessary parameterizations or redo the initialization, as mentioned later in this paragraph. We make sure to use the exact restart (rewriting parameterizations with free parameters so they resemble the main initiliazation) as much as possible to eliminate the effect of a different initialized state as much as possible

Line 396: Relative mass loss to what?

Compared to the initialized state, this will be added

Line 401: I disagree, as your melting experiments show that the timescales of retreat depend also on the melt rate applied. They can hence not only be controlled by bed topography or internal system dynamics.

Yes, the timescale of the start of the 'fast phase'. The 'fast phase' itself however, progresses roughly the same way in all simulations (the 250 year steep part in the curve). We will clarify this

Table 3: Can you add the maximum and average rates of ice loss during "collapse"?.

We thank the reviewer for this great suggestion, and we will add those to Table 3.

Line 425: Please see introduction about how you can compare these studies. It would be interesting to also compare with Feldmann C Levermann 2015, Reese et al., 2023 mass loss rates here, and analyze differences.

We will rewrite the introduction (mainly targeting the framing of our method and the literature review), and change this part accordingly

How much does the choice of the dataset of rates of ice thickness change influence your results?

We did not assess this, because we did not have another mass change rates dataset with this spatial coverage at our disposal. We tried to lower the mass change rates with 25%, and we did not see a large influence. We also decreased our SMB and basal melt rates with certain percentages during the run and did see a slowdown but not a prevention of the collapse (which we choose to be the main point of our study).

How much would variability in ocean forcing affect the timescales?

We think this will have a moderate effect on the timescales, since

changing the basal melt rates with 25% or 50% siginificantly slows down the collapse

 A discussion of caveats is missing. This will be incorporated in the manuscript in section 'Missing processes'.

**6. Conclusions**

As mentioned before, your method is not particularly new. See earlier replies

We will change this in the introduction and we will also change this here.

Line 461: Add here your maximum timescale of collapse, and you should mention that this really depends on modelling choices. Great suggestion, we will add this

Line 464: Except for your lower melt experiments where the phase appears to be longer.

We note a slight increase in collapsing timescales, and we will add this as nuance to the conclusion

Line 465: What means "irreversible for current climate"? Irreversible is a concept from non-linear dynamics, are you applying this here? Irreversible in the sense that the current configuration of TG and PIG cannot co-exist with the current ocean/atmosphere.

We could not test if applying the negative of the perturbation that caused the mass change rates will regrow the glacier, so we will remove the word and claim 'irreversible'

Line 469: I think you want to consider natural variability in ocean conditions and how that influences your argument.
Great suggestion, we will add this

**Data and code availability:**

Consider publishing your scripts and model outputs on a repository such as zenodo. We will do this

---

## Author Response (AR2)

**Minor revisions rebuttal 'Present-day mass loss rates are a precursor for West Antarctica Ice Sheet collapse'.**

Replies are written in blue

**Reviewer 1**

This is my second review of the manuscript "Present-day mass loss rates are a precursor for West Antarctic Ice Sheet collapse" by T. van den Akker and colleagues and I want to thank the authors for the detailed response and extensive revisions to the manuscript. I find this new version to be a lot more clear and easier to follow, which significantly strengthens the conclusions. The changes respond to most of the comments suggested and the new version is a lot more accurate. I mostly have minor comments listed below. The two things to be careful about is to add more discussion about the calving and be clear about the sliding and friction laws used, as there seems to be some mixed up here.

We thank the reviewer for their second read of our manuscript and the detailed comments and suggestions below. The line numbers in the reviewers comment were off, so in some cases it was hard to find which sentence the readers was referring to. We added the new line numbers in the corrected manuscript in our answers below.

Below are some detailed comments, line numbers refer to the manuscript with track-changes.
l.14: this should mention both global and local scales and not just local scales we will change this to 'a global and local scale'
l.18: "physics" -> "physical" we will change this
l.27: "Dynamical processes": be more specific or at least mention that it is something different from the current state (because they are already dynamic things ongoing) we will change this to 'the resulting GMSL from non linear processes'
l.56: remove "dynamical" We will rewrite this sentence according to the suggestion of reviewer 2
l.290: "with a regularized" -> "and a regularized" This was not mentioned at line 290 but at 280, and we will change it it as 'as a regularized'
l.289-295: Discuss also calving in this paragraph. We will add the sentence 'The calving front is maintained at its present day position. This conservative approach has been chosen because calving physics as well as the impact of ice berg melange of ice shelves are still poorly understood, and overly retreating calving fronts enhance mass loss which make the results overly negative.'
Table 1: put "inverted quantity in this study" in parenthesis both times we will do this
l.331: "It is based" -> " It depends" or "It is a function of" This was found at line 158, and we will change this to 'It is a function of'
l.340: "with linear" -> "with a linear" Ln 160, and we will change this accordingly
l.358: "not represented" -> "not included" Ln 178, and we will change this to 'not included'
l.383: "assumed" -> "assume" Ln 185: we will remove the 'd'
l.388: "with increasing" -> "leading to increased" Ln 191, we will change this

l.397: I think what is said here is about the water column thickness in the cavity. Ln 199: correct, we will change this to 'the water column thickness in the cavity'

l.410: "it thins" -> "the ice thins" Ln: 213. We will change this accordingly

l.486: "by model drift" -> "by model drift from the initialization" Ln 234: we will add this suggestion.

l.559: "against to an extensive" -> "against an extensive" Ln 276: we will remove 'to'

l.561: "from collapsing PIG" -> "from collapsing" ln 278: we will remove 'PIG'

l.616: Budd and regularized Coulomb are two different sliding laws, so this is not accurate (it looks like Budd sliding according to Eq. 1.10). Ln 295: We will remove the part between parenthesis

l.750: Provide the RMSE number to facilitate comparison. We assume this is about Ln 334, we will add the velocity RMSE of 156 m/yr

Fig.2: Given the very similar patters at year 1 and 5, and would suggest remove one of these two years and instead adding a similar figure for the UFEMISM model as the third subplot. We would like to highlight with this figure that CISM does not get the dh/dt pattern as initial 'shock' which dissappears in a few timesteps, but is actually able to have this pattern persist. Adding the UFEMISM dh/dt here would disctract from that. We will add to Ln 370: 'The overal pattern of the modelled mass change rates changes little during the first timesteps of a continuation simulation, highlighting that CISM is able to exhibit the observed mass change rates robustly'

Figure 4 caption: I think the location is shown on figure 3 Correct, we will change this and remove 'in the main text'

l.981: "The ice thickness change shown in Figure 7" ... it would be great to remind over which period these thickness changes happen. We will add 'after 1000 years of unfocred forward run'

Figure 7 caption: "The original" -> "The initial" we will change this

l.1011: the sensitivity of what? Ln 505: We will rewrite this sentence to 'To assess the sensitivity of our results to those choices and parameter values we varied them within the range we viewed as physically realistic.'

l.1019: "relative mass loss and resulting GMSL rise relative o the initial state": a few different initial states are used for some experiments (when some ice parameters are changed and a new spin-up is required), so it would be good to clarify which initial state is used (always the default one or the one corresponding to each experiment). I suspect it is the later, so what is the impact of using slightly different initial states? It is the latter, and the impact was small to neglible, we verified that the difference in terms of veloocity, thickness and grounding line position with the original spinup was small and we performed model drift experiments to be sure the WAIS did not collapse using these slightly altered initial states. We will add to line 1019: 'For some simulations, marked with an '*' in Table 3, a new initialized stated was necessary. We made sure that these new initial states were close to the default initial state in terms of thickness and velocity RMSE and grounding line position. We performed the same model drift experiments to verify that for these new initialized states, the WAIS would not collapse without adding the present-day mass change rates. '

l.1026: I would not call that a fast phase but rather a collapse or fast retreat phase. Ln 520: We will adopt 'Fast retreat phase'

l.1066: "with about" -> "by about" Ln 543: we will add this

Figure 8 caption: "resulting" -> "equivalent" and "reach 50%" -> "need to reach 50%"

We will add this
l.1156-1157: "Previous initialization methods .... Compared to data assimilation initialization methods." This sentence is rather unclear. First what is the difference between "previous initialization methods" and "data assimilation initialization methods" and second the data assimilation initialization methods are also not able to capture mass change rates in most cases. So this part is really confusion and should be removed or clarified. We agree that this sentence is confusing. We will rewrite this sentence to 'Previous initialization methods, both the spinup and data-assimilation initialization method, were unable to accurately represent the observed mas change rates.'
l.1209: should be section 8 (not 7) We note the error and will change this to 8
l.1212: mention under present-day conditions. Ln 625: we will add this
l.1219: The last sentence is kind of a very weird way to conclude the paper, I would not end with these words. We agree, and will remove this sentence

**Reviewer 2**
Dear van den Akker et al.,
Thanks for including the comments, very much appreciated. The manuscript has substantially improved and I am happy with it to be published subject to minor corrections outlined below.

We thank the reviewer for the second read, and the usefull, detailed and constructive comments below. Our answers are given in blue

Reply on comment on abstract, line 13 (reviewer 2): To derive the statement from Aschwanden et al., 2019 that models struggle to reproduce present-day rates of ice loss is unfair because models were never asked to try and reproduce present-day ice loss in ISMIP6 project (which the Aschwanden study is based on). The initialisation was left to the modellers. Hence, this does not show that models cannot reproduce trends. A valid conclusion that can be drawn from Aschwanden et al. and other studies is that "Recent studies highlighted the need for ice sheet models to be benchmarked against reproducing present-day, observed mass change rates." I suggest you simply change the sentence to this statement. Thanks for this constructive point, we will change Ln 13 as suggested

Line 31, new manuscript: definition of "dynamical irreversible retreat (i.e., the recovery time is substantially longer than the response time)" – not only that, but you have hysteresis behavior. Once retreated, the system is not expected to recover even if the climate forcing is reduced below the pre-collapse value – in your response this was stated as "i.e. the glacier will not return to its present grounding line position when the forcing is removed". Maybe go back to this?
We will change this back to the response of the first rebuttal, as suggested

Line 41, new manuscript: "model studies" neither Jenkins et al., 2016 not Thoma et al. 2008 model the response of PIG ice dynamics to CDW intrusions (none employs an ice sheet model). We will remove the word 'model'

Line 52, new manuscript: "while others suggest that the two glaciers will require additional forcing to collapse (Feldmann and Levermann, 2015; Arthern and Williams, 2017; Reese et al., 2023; Garbe et al., 2020)". This seems to be a misunderstanding of the authors of these studies: neither Arthern & Williams nor Feldmann & Levermann make statements that additional forcing for collapse is needed. Reese et al., 2023 even finds collapse under current climate in line with the studies cited in the sentence before, Garbe et al., identifies the tipping point between 1 and 2 degree above pre-industrial levels, which again includes the current climate state. Reformulate to "A number of recent studies suggest that TG and PIG are unstable under the current climate and could collapse on a timescale up to 2000 years (Golledge et al., 2021; Coulon et al., 2024; Reese et al., 2023; Garbe et al., 2020)." This will impact the following sentences as well.

We will rewrite this to: 'A number of recent studies suggest that TG and PIG are unstable under the current climate and could collapse on a timescale up to 2000 years (Garbe et al., 2020; Golledge et al., 2021; Reese et al., 2023; Coulon et al., 2024). Lipscomb et al. (2021) projected accelerated retreat leading to a collapse only when ocean thermal forcing increases by 1–2 K relative to a preindustrial or 20th century equilibrium. However, there are few historical observations of Southern Ocean temperatures, so it is unknown how much the ASE has warmed in the past century, and if the warming needed to drive such a retreat has already happened.'

Line 63, new manuscript: "These models often struggle to represent the observed mass change of recent decades, because the result of a steady state initialization is a stable ice sheet without model drift" – instead I would say "Since they are initialised to a stable ice sheet without model drift, they do not represent the observed mass change of recent decades." We agree to this comment and will replace the mentioned text by the suggestion

Reply on comment on Line 85 (reviewer 2): Sorry for the jargon, with PSU I meant the Penn State Model developed by Dave Pollard. Force-to-thickness I meant a flux correction as described in Aschwanden et al., 2016 "Complex Greenland outlet glacier flow captured." We thank the reviewer for this clarification

New figure 1: Panel (d) shows several locations along Thwaites grounding line where your inversion derives "zero" melt, is that correct? That is correct. These are locations where CISM underestimates the ice thickness, and the basal melt scheme tries to compensate that by lowering the basal melt rates. Basal melt rates below zero are not represented in CISM.

New figure 2: This is a CISM run, right? Figure 2a are the observations, figure 2b and c are the mass change rates modelled by CISM after 1 and 5 years of running forward. We will add to the title 'Observed and modelled present-day mass change rates'

New line 448: Holland et al., 2019, did not model things, but analyse model results. We will remove 'model'

New figure 7: Change "future state" in the caption. We do not see a fitting alternative, and we would like to keep the caption as close to the caption of Figure 3 as possible

New line 611: You could reformulate to "Replication of our experiments with other ice sheet models could substantiate whether the collapse of Thwaites and Pine Island Glaciers is indeed unavoidable without a considerable reduction of basal melt as indicated in our study." and remove the last sentence. We agree to this suggestion and will remove the last sentence and change the second to last sentence as suggested.